# Adenosine triggers early astrocyte reactivity that provokes microglial responses and drives the pathogenesis of sepsis-associated encephalopathy in mice

Qilin Guo [1,2], Davide Gobbo [1], Na Zhao [1,3], Hong Zhang[4], Nana-Oye Awuku [5], Qing Liu[1], Li-Pao Fang [1,2], Tanja M. Gampfer[6], Markus R. Meyer[6], Renping Zhao [4], Xianshu Bai [1,2], Shan Bian[7], Anja Scheller [1,2], Frank Kirchhoff [1,2] ✉ & Wenhui Huang [1,2] ✉

Molecular pathways mediating systemic inflammation entering the brain parenchyma to induce sepsis-associated encephalopathy (SAE) remain elusive. Here, we report that in mice during the first 6 hours of peripheral lipopolysaccharide (LPS)-evoked systemic inflammation (6 hpi), the plasma level of adenosine quickly increased and enhanced the tone of central extracellular adenosine which then provoked neuroinflammation by triggering early astrocyte reactivity. Specific ablation of astrocytic Gi protein-coupled A1 adenosine receptors (A1ARs) prevented this early reactivity and reduced the levels of inflammatory factors (e.g., CCL2, CCL5, and CXCL1) in astrocytes, thereby alleviating microglial reaction, ameliorating blood-brain barrier disruption, peripheral immune cell infiltration, neuronal dysfunction, and depression-like behaviour in the mice. Chemogenetic stimulation of Gi signaling in A1AR-deficent astrocytes at 2 and 4 hpi of LPS injection could restore neuroinflammation and depression-like behaviour, highlighting astrocytes rather than microglia as early drivers of neuroinflammation. Our results identify early astrocyte reactivity towards peripheral and central levels of adenosine as an important pathway driving SAE and highlight the potential of targeting A1ARs for therapeutic intervention.

Severe systemic inflammation caused by infections or injuries can eventually deteriorate to a life-threatening status called sepsis[1]. Such systemic inflammation also induces impaired functions of the central nervous system (CNS), one of the first affected organs in sepsis, leading to so-called sepsis-associated encephalopathy (SAE)[2]. Neuroinflammation is a hallmark of SAE, contributing to blood-brain barrier (BBB) damage, altered neuronal activity, and sickness/depression-like behaviour[3,4]. Immune signals mediating systemic inflammation were shown to evoke rapid inflammatory responses firstly in components of

the BBB (i.e., endothelial cells and pericytes of the brain vasculature) 1-2 h after the systemic inflammation challenge, subsequently releasing inflammatory mediators such as prostaglandin E2 or the chemokine (C-C motif) ligand 2 (CCL2, also referred to as monocyte chemoattractant protein 1, MCP1) and further potentiating neuroinflammation as well as aberrant neuronal activity[3,5–7]. However, astrocytes were usually considered as targets of reactive microglia after a systemic inflammation challenge rather than early drivers of neuroinflammation themselves, which is counterintuitive concerning their

anatomic positions as gatekeepers of the BBB easily receiving signals from the circulation.

Previous studies focused on augmented proinflammatory cytokines (e.g., TNFα, IL-1α and β, and IL-6) of the blood as immune signals which induce inflammatory responses in the CNS[3]. In addition to cytokines, different purine metabolites such as ATP and adenosine can be found at elevated blood levels after a systemic inflammation challenge[8–11]. However, the roles of purinergic signaling, if any, in affecting initiation and progression of SAE are largely undetermined. In the present study, we examined whether and how extracellular adenosine, the end-product of ATP hydrolysis, could regulate neuroinflammation using a well-established mouse model of SAE in which sepsis is induced by peripheral (intraperitoneal, i.p.) injection of the endotoxin lipopolysaccharide (LPS)[5,12–14]. We detected rapid increases of extracellular adenosine levels in the blood and the brain shortly after the LPS injection which were followed by neuroinflammatory response of astrocytes, microglia, and neurons in the CNS. We further showed the first in vivo evidence that elevated plasma adenosine signal could pass the BBB to enhance the central adenosine levels. Direct i.p. injection of adenosine or its analogues (i.e., 5′-N-ethylcarboxamide adenosine, NECA; and N6-cyclopentyladenosine, CPA) could induce an upregulation of several proinflammatory factors in the brain, which could be prevented by specific ablation of adenosine A1 receptors (A1ARs) in astrocytes rather than in pericytes, oligodendrocyte precursor cells or microglia, suggesting adenosine as an inflammatory mediator between the body and the CNS via triggering astrocyte reactivity. Comprehensive analyses of mice with inducible and astrocyte-specific A1AR conditional knockout (cKO) mice in the LPS-induced sepsis model further revealed that adenosine triggers astrocyte reactivity via A1ARs which provokes the inflammatory response of microglia at the early phase of systemic inflammation, driving the following pathogenesis of SAE by disrupting the BBB integrity, generating neuronal dysfunction as well as depression-like behaviour of the septic mice.

So far four subtypes of adenosine receptors (i.e., A1, A2a, A2b, A3) have been discovered. However, most previous studies characterizing the neuroimmunological functions of adenosine focused on A2 and A3 AR signaling. Still, our knowledge on the role of A1ARs in regulating neuroinflammation remained scarce[15–17]. Here, we unveil an important pathway employing adenosine as signaling molecule that mediates systemic inflammation to induce neuroinflammation by triggering early astrocyte reactivity via A1AR signaling. Moreover, we provide evidence that early reactive astrocytes act as drivers of neuroinflammation rather than being the targets of reactive microglia in SAE.

## Results

### Systemic inflammation increases adenosine levels in the blood and brain

Peripheral administration of a high dose of LPS (such as 5 mg/kg, therefore 1 mg/kg is referred to as low dose in this study) is a widely used method to induce septic systemic inflammation in mice[5,12,14]. Sepsis is known to increase plasma adenosine levels in patients or volunteers injected with low doses of LPS[8,10]. In the mouse model, we also found elevated levels of plasma adenosine at 2 hours post injection (hpi) of LPS, which peaked at 6 hpi, and subsequently dropped to baseline level at 12 and 24 hpi (Fig. 1a, b). Previous studies suggested that systemic inflammation could disrupt BBB integrity[4,18] and adenosine analogues acting on endothelial A1 and A2a ARs result in the opening of the BBB[19]. We also observed that peripheral administration of adenosine itself could increase BBB permeability as assessed by Evans blue (EB) extravasation experiments (Fig. 1c). To further correlate adenosine increase with BBB disruption in systemic inflammation, we performed EB extravasation experiments to study the dynamic changes of BBB integrity after LPS injection. Mice were injected i.v. with EB two hours prior to each analysis time point (Fig. 1a).

Intriguingly, our experiments detected BBB disruption already at 2 hpi, which, similar to adenosine changes in the blood, reached the peak at 6 hpi and decreased at 12 and 24 hpi (Fig. 1d). Furthermore, it has been shown that a peripheral injection of LPS could activate various glial cells and pericytes as well as several cytokines in the brain within 24 hours[5,6,14]. In our experiments, we also observed enhanced expression of genes related to inflammation (e.g., *Ccl2, Ccl5, Cxcl1, Cxcl10, Tnf, Il6, Il1a*, and *Il1b*), astrocyte (*Gfap* and *Lcn2*) or microglia reaction (*Itgam*) in the mouse cortex. Peaks of expression of the inflammation-related proteins reached between 2 and 6 hours post injection (Supplementary Fig. 1a–c), concomitant with a plasma adenosine increase and a BBB disruption, as signs of systemic inflammation.

To investigate whether systemic inflammation could also evoke extracellular adenosine level in the brain parenchyma, we performed in vivo 2-photon laser-scanning microscopy (2P-LSM) of a novel genetically encoded G protein-coupled receptor (GPCR)-activation-based (GRAB) sensor for adenosine (GRAB_{Ado1.0}) (Fig. 1e)[20] expressed in astrocytes in the somatosensory cortex. We detected increased fluorescence intensities (F.I.) of GRAB_{Ado1.0} already at 2 hpi, reaching plateau levels at 6 hpi (Fig. 1f, g), suggesting that LPS-induced systemic inflammation could increase the extracellular adenosine level in the brain, in agreement with prior studies using intracerebral biosensor recordings in rat[21]. Next, we examined whether peripheral adenosine in the blood could directly induce elevation of extracellular adenosine level in the brain. We injected i.p. adenosine (20 mg/kg) supplemented with sulforhodamine 101 (SR101, red fluorescence) to GRAB_{Ado1.0}-expressing mice during 2P-LSM live imaging (Fig. 1h; Supplementary Movie 2). We were able to detect increased GRAB_{Ado1.0} F.I. after the adenosine injection, which is concomitant with the increase of SR101 F.I. in the blood vessels, while injection of SR101 alone could not increase the GRAB_{Ado1.0} F.I. (Fig. 1i, j; Supplementary Movie 1). In addition, we observed a dose-dependent increase of GRAB_{Ado1.0} F.I. when we injected different doses of adenosine (5–20 mg/kg, Fig. 1k), indicating that increased peripheral adenosine can directly enhance the extracellular adenosine level in the brain. Taken together, we determined the dynamic changes of adenosine levels in the blood and somatosensory cortex post-peripheral LPS challenge. Although the potential sources of the increased extracellular adenosine in the brain after LPS treatment remain unidentified and there is a lack of evidence from previous studies showing adenosine could pass the BBB under normal conditions, our current results strongly suggest that peripheral LPS injection induced a rapid increase of plasma adenosine which may contribute to the elevated extracellular adenosine level in the brain as well as a rapid neuroinflammatory response in the early phase of the systemic inflammation with a disruption of BBB function.

### Peripheral adenosine administration induces astrocyte reactivity and neuroinflammation via A1ARs

To evaluate if adenosine could evoke a neuroinflammatory response, we injected adenosine (i.p., 5 mg/kg, six times, at one-hour interval) to 'wild-type' (wt) control (ctl) mice and analysed gene expression levels in the mouse cerebral cortex by qPCR at 6 h post the first injection (Fig. 2a). We injected adenosine several times due to its short lifetime in the blood (~1 h)[22]. Intriguingly, *Lcn2* (a marker for inflammatory reactive astrocytes)[23] was upregulated (Fig. 2b; Supplementary Fig. 2a) upon adenosine administration. Moreover, several other proinflammatory cytokine/chemokine genes such as, *Ccl2, Cxcl1, Il1a* and *Tnf* also rapidly responded to the adenosine application with increased expression level (Fig. 2b; Supplementary Fig. 2a). Because the plasma adenosine level increased rapidly with the LPS injection and peaked at 6 hpi, we injected NECA (1 mg/kg, a non-selective adenosine analogue, half-life = ~5 h)[19,24] to ctl mice and analysed cytokine expression at 6 hpi (Fig. 2a). We observed that several cytokines such as *Cxcl1, Cxcl10, Ccl2, Tnf, Il6* as well as *Lcn2* were also upregulated upon NECA treatment (Fig. 2b; Supplementary Fig. 2b). In addition, we injected CPA (1 mg/kg,

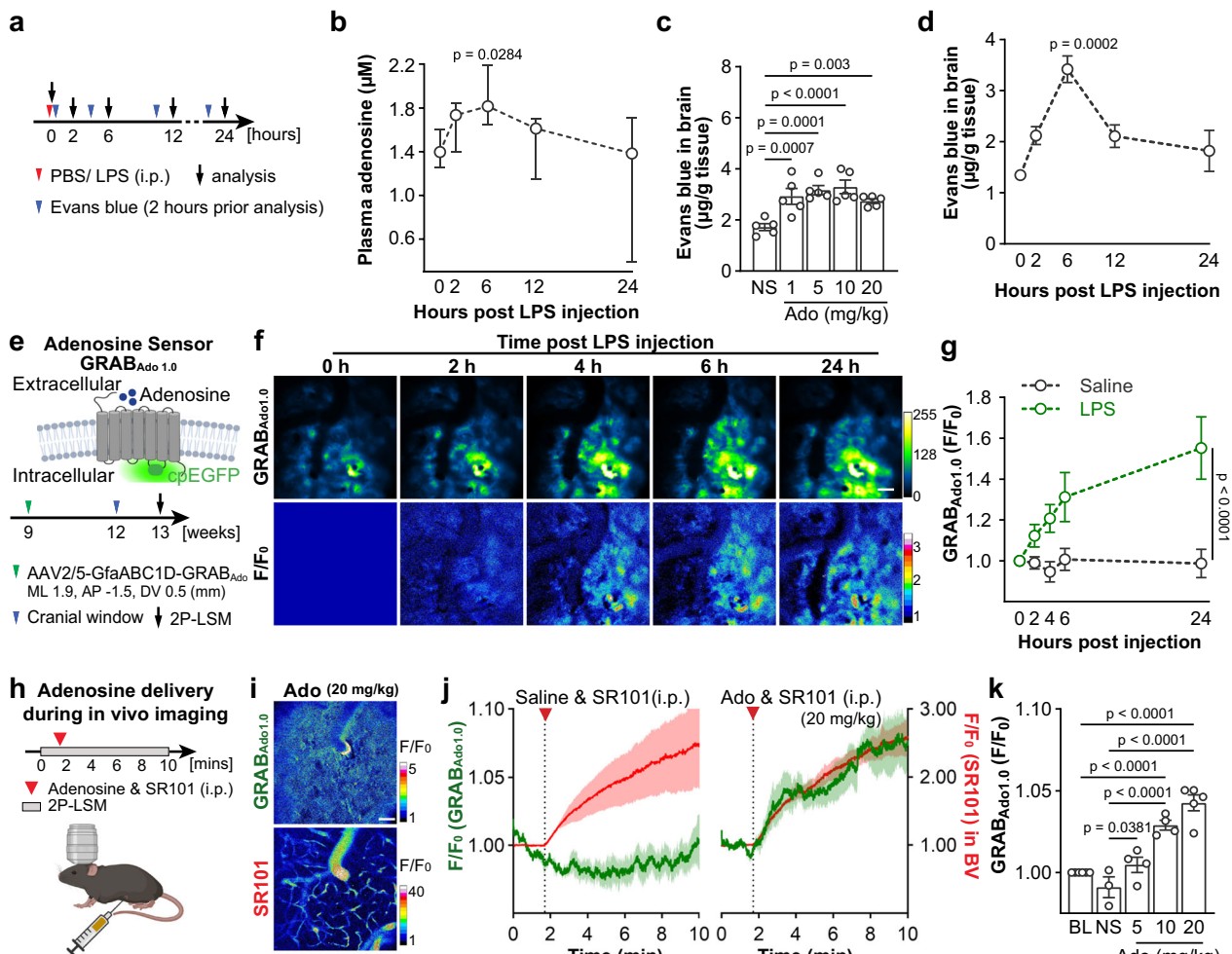

**Fig. 1 | Peripheral LPS challenges increase adenosine levels in the blood and brain. a** Schematic illustration of the experimental setup and time course for analysis of plasma adenosine levels and Evans blue (EB) injection. **b** Plasma adenosine concentration rapidly increased post peripheral LPS injection (n = 5 mice for each time point). **c** EB extravasations increased in the brain after peripheral adenosine injection. (n = 5 mice for each time point). NS: saline; Ado: adenosine. **d** Peripheral LPS injection increased EB extravasations in the mouse brain (n = 3 mice for each time point). **e** Schematic illustration of the principle of the GRAB$_{Ado1.0}$ sensors (up) and cortical extracellular adenosine level measurement post LPS injection by using GRAB$_{Ado1.0}$ in vivo 2P-LSM live imaging (down). GfaABC1D-GRAB$_{Ado1.0}$ plasmids were delivered to cortical astrocytes via AAV2/5 injection according to the coordinates indicated in the scheme. **f** Representative fluorescence images (up) and pseudocolor images (down) of GRAB$_{Ado1.0}$ signals post peripheral LPS injection. Scale bar = 50 μm. **g** Comparison of relative fluorescence intensities (F.I.) of GRAB$_{Ado1.0}$ acquired in the LPS/Saline injection model. The recording of 0 h after LPS/Saline injection was used as F$_0$. (n = 3 mice per group). **h** Schematic illustration of injection of adenosine supplemented with

SR101 (i.p.) during the in vivo imaging of GRAB$_{Ado1.0}$. **i** Representative pseudocolor images of GRAB$_{Ado1.0}$ signal and SR101 signal after a peripheral adenosine and SR101 injection. Scale bar = 50 μm. (n = 5 mice). **j** Increase of GRAB$_{Ado1.0}$ signal (green) after the injection of adenosine (20 mg/kg, i.p.) was concomitant with the increase of SR101 signal (red) in blood vessels (BV), while GRAB$_{ado1.0}$ signal is not altered after saline injection. **k** Relative fluorescence intensities (F.I.) of GRAB$_{Ado1.0}$ upon applications of various dosages of adenosine. The baseline was used as F$_0$. (baseline n = 6 mice, saline n = 3 mice, ado 5 mg/kg n = 4 mice, ado 10 mg/kg n = 4 mice, ado 20 mg/kg n = 5 mice). BL: baseline; NS: saline; Ado: adenosine. Summary data are presented as mean ± SEM in **c**, **d**, **g**, **j**, **k**, and as median ± IQR in **b**. Statistical significance in **b** was assessed using Kruskal–Wallis test uncorrected Dunn's test; statistical significance in **c**, **k** were assessed by one-way ANOVA, Fisher's LSD test; statistical significance in **g** was assessed by two-way ANOVA, Fisher's LSD test. Source data are provided as a Source Data file. Panels **e**, **h** was created with BioRender.com released under a Creative Commons Attribution-NonCommercial-NoDerivs 4.0 International license.

a selective A1AR agonist, half-life = ~0.5 h) to ctl mice and also detected upregulated cytokines and other proteins related to inflammation/astrocyte activation at 6 hpi (Fig. 2a, b; Supplementary Fig. 2c). Furthermore, when we combined the injection of CPA with a low dose of LPS (1 mg/kg, LPS$^{low}$), the expression levels of all tested cytokines or chemokines were strongly augmented compared to mice injected with LPS only (Fig. 2c, Supplementary Fig. 2f). On the other hand, in the cortex of high-dose (5 mg/kg, LPS$^{high}$) LPS-challenged mice the administration of DPCPX (1 mg/kg, 8-Cyclopentyl-1,3-dipropylxanthine, a selective A1AR antagonist) inhibited the activation of glial cells and neurons (Supplementary Fig. 2d, e) as well as reduced the expression of several inflammatory factors (e.g., *Ccl2, Tnf, Lcn2, Il6,*

and *Cxcl10*) (Fig. 2c; Supplementary Fig. 2g). In conclusion, our results strongly suggest that A1AR signaling directly contributes to the induction of neuroinflammation.

Recent transcriptomic studies suggested that A1ARs are the most abundantly expressed ARs in astrocytes[25,26]. To determine the contribution of astrocyte-specific A1ARs to adenosine-evoked neuroinflammation, we crossbred GLAST-CreERT2 mice with temporal control of astrocyte-specific gene recombination[27] to floxed *Adora1* mice[28], thereby generating inducible astrocytic *Adora1* cKO mice (Fig. 2d) (for simplicity they were termed *Adora1* cKO mice afterwards). In addition, we crossbred RiboTag mice[29] enabling specific and direct purification of mRNAs (mRNA$^{RiboTag}$) from Cre-expressing cells in cerebral cortices

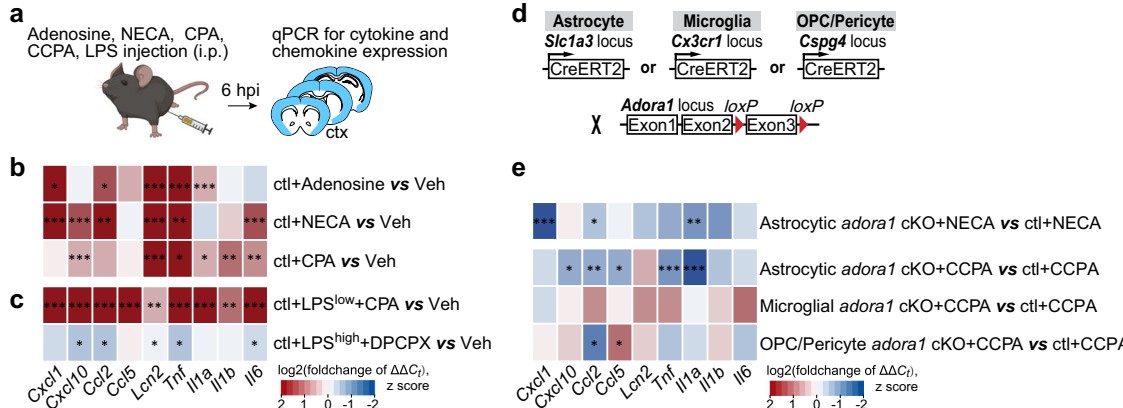

**Fig. 2 | Peripheral adenosine administration evokes upregulation of inflammation-related genes in the brain. a** Schematic illustration of adenosine, adenosine analogue (NECA), A1 adenosine receptor (A1AR) agonists (CPA, CCPA), and A1AR antagonist (DPCPX) administration experiments. **b** Expression of inflammation-related genes was enhanced in the mouse cortex six hours post adenosine, NECA, and CPA injections (n = 3 mice per group). **c** CPA further upregulated the inflammation-related genes in the cortex induced by a peripheral LPS[low] (1 mg/kg, i.p.) injection. (n = 3 mice per group). However, DPCPX administration reduced the inflammation-related genes in the cortex induced by a peripheral LPS[high] (5 mg/kg, i.p.) injection. (n = 3 mice per group). **d** Schematic illustration of mouse breeding. **e** Inflammation-related gene expressions were reduced in the

cortex of astrocytic *Adora1* cKO mice at 6 hours post NECA (n = 3 mice per group) and CCPA (n = 6 mice per group) injection compared to ctl mice, which was not observed in mice with specific ablation of *Adora1* in microglia (using Cx3CR1-CreERT2 mice) and oligodendrocyte precursor cells/pericytes (using NG2-CreERT2 mice, only *Ccl2* was reduced while *Ccl5* increased). Summary data are presented as the log2(foldchange of ΔΔCt). Statistical significance of each gene expression in **b**, **c**, **e** was assessed to ctl+veh using two tailed unpaired Student's t test. Bar graphs of **b**, **c**, **e** was presented in Supplementary Fig. 2a–i. Source data and exact *P* values are provided in the Source Data file. Panel **a** was created with BioRender.com released under a Creative Commons Attribution-NonCommercial-NoDerivs 4.0 International license.

to avoid pitfalls inherent to cell-sorting procedures (Supplementary Fig. 3a). We performed qPCR and RNA-seq analysis for *Adora1*[RiboTag] expression to reveal a successful gene excision of *Adora1* in cKO mice (Supplementary Fig. 3a–d). In parallel, we used a genetically encoded Ca²⁺ indicator mouse line (Rosa26-LSL-GCaMP3, Rosa26-GCaMP3)[30] to study Ca²⁺ activity in astrocytes after tamoxifen administration (Supplementary Fig. 4a). We recorded cortical astrocytic Ca²⁺ signalling from astrocytic *Adora1* cKO and ctl mice using ex vivo 2P-LSM before and after the focal application of CPA (1 μM) in the presence of the neuronal activity blocker tetrodotoxin (TTX, 1 μM) (Supplementary Fig. 4b, c). The automatic detection of regions-of-activity (ROAs) was performed as previously described[31] using the custom-made MATLAB-based tool MSparkles[32]) (Supplementary Fig. 4c–e). To note, genetic ablation of astrocytic A1ARs was associated with altered baseline Ca²⁺ activity, including an increased number of Ca²⁺ signals with lower amplitude. In particular, A1AR-deficient astrocytes showed increased occurrence of spatially-defined ROAs with reduced area but smaller signal frequency per ROA (Supplementary Fig. 4f, g). Taken together, these results showed that the removal of astrocytic A1ARs results in the reduction of astrocytic Ca²⁺ signals ex vivo in terms of signal amplitude, spatial extension and signal frequency. On the other hand, CPA application triggered significant increases of astrocytic Ca²⁺ signals in ctl mice which was not detected in astrocytic *Adora1* cKO mice, further confirming the functional removal of A1ARs from astrocytes.

Subsequently, we injected NECA or CCPA (0.3 mg/kg, 2-chloro-N6-cyclopentyladenosine, a more selective A1AR agonist) to *Adora1* cKO and ctl mice and found that the expression of several upregulated cytokines and chemokines (e.g., *Ccl2, Ccl5, Cxcl1, Cxcl10, Il1a, and Tnf*) was significantly attenuated in *Adora1* cKO mice (Fig. 2e, Supplementary Fig. 2h), which was not observed in mice with specific ablation of *Adora1* in microglia (using Cx3CR1-CreERT2 mice)and oligodendrocyte precursor cells/pericytes (using NG2-CreERT2 mice, only Ccl2 was reduced) (Fig. 2d, e; Supplementary Fig. 2j, k). In line with the in vivo observation, application of CCPA to primary astrocytes evoked significant upregulation of several chemokines (e.g., *Cxcl1, Cxcl10, Ccl5*), whereas CV1808 (a non-selective antagonist of A2AR) did not cause significant expression alterations of all the tested inflammation-

related genes (Supplementary Fig. 5a–c). Altogether, these results provided additional evidence that adenosine could trigger neuroinflammatory response via astrocytic A1AR signaling.

## A1AR signaling augments inflammatory response of reactive astrocytes in the early phase of systemic inflammation

Upregulation of the immediate early gene *Fos* family is frequently used to indicate activation of neurons as well as early reactive astrocytes upon pathological stimulation[33–37]. Furthermore, c-Fos expression can be triggered by GPCR signaling in astrocytes. Therefore, we firstly performed immunohistochemistry of c-Fos combined with the astrocyte marker Sox9 on mouse brain slices to investigate the time course of astrocyte reactivity after LPS (5 mg/kg, and for all the following experiments) injection. We found that the expression of c-Fos in astrocytes of ctl mice was unaltered at 2 hpi but was upregulated at 6 hpi and reduced to basal level at 24 hpi. However, the c-Fos expression in astrocytes in cortex, hippocampus, and striatum was significantly inhibited by the deficiency of A1ARs at 6 hpi, indicating a critical early temporal window of the astrocytic response to a systemic inflammation (Fig. 3a–d). In addition, activation of signal transducer and activator of transcription 3 (STAT3) signaling is an important inducer of inflammatory responses of reactive astrocytes[38]). To evaluate the astrocyte inflammatory response, we performed immunohistochemistry for the activated form of STAT3 (i.e., phospho-STAT3[Tyr705], p-STAT3). In the brain of ctl and *Adora1* cKO mice, we observed that p-STAT3 was almost non-detectable at 0 hpi. However, at 6 hpi the p-STAT3 immunoreactivity could be detected in virtually all Sox9⁺ astrocytes of ctl mice which was significantly reduced in *Adora1* cKO mice. At 24 hpi, the p-STAT3 expression level was largely reduced (Fig. 3e–h). These results further suggest that the activation of A1AR signaling promote the inflammatory response of reactive astrocytes which peaks in the early phase of the systemic inflammation.

Next, we performed high-throughput RNA-seq to further study the impact of A1AR-mediated reactive astrocyte response to systemic inflammation at molecular level. We used the RiboTag approach to directly purify astrocytic translated mRNA (mRNA[RiboTag]) from the

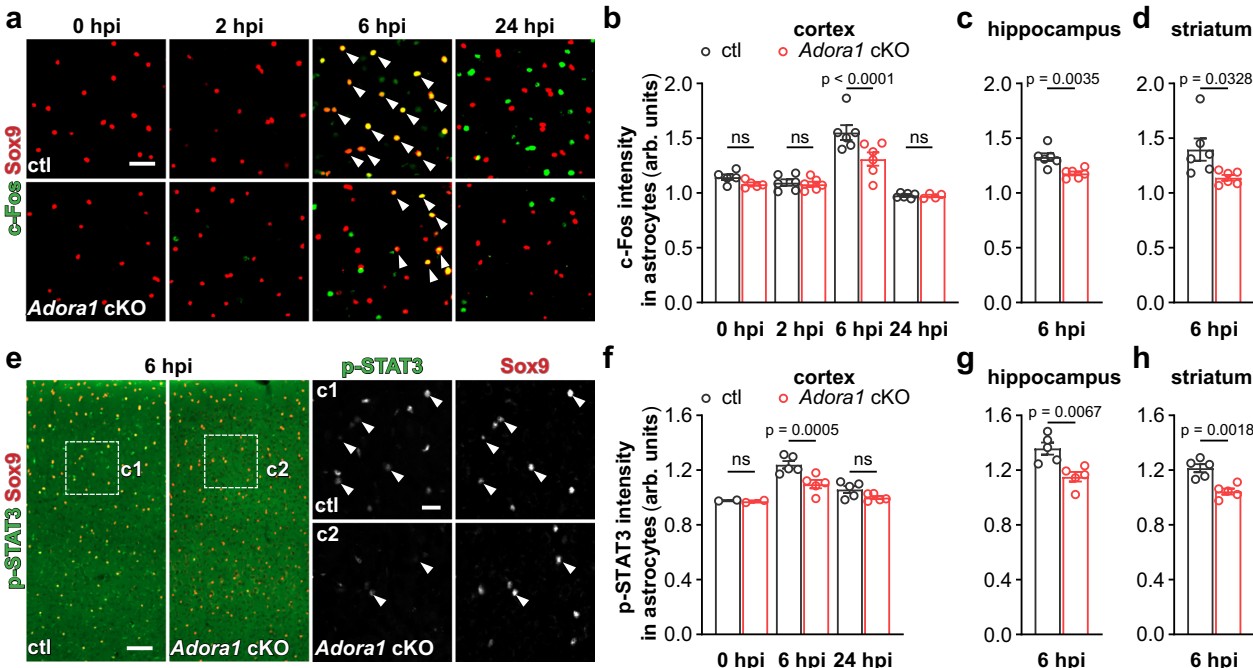

**Fig. 3 | A1AR-deficient astrocytes are less reactive to the peripheral LPS challenge. a** Representative images of c-Fos expression in cortical Sox9+ astrocytes (arrowheads) post LPS injection. Scale bar = 20 μm. **b–d** Astrocytic c-Fos immunofluorescence intensity (arbitrary unit, arb. units) was enhanced at 6 hpi in the cortex, hippocampus, and stratum of ctl mice which was inhibited in *Adora1* cKO mice (n = 5 mice in ctl and *Adora1* cKO at 0 hpi, n = 6 mice in ctl and *Adora1* cKO at 2 hpi and 6 hpi, n = 6 mice in ctl at 24 hpi, n = 4 mice in *Adora1* cKO at 24 hpi). **e** Representative images of p-STAT3 expression in cortical Sox9+ astrocytes

(arrowheads) post LPS injection. Scale bar = 50 μm in **e** and 20 μm in **c1, c2**. **f–h** Astrocytic p-STAT3 immunofluorescence intensity (arb. units) was enhanced at 6 hpi in the cortex, hippocampus, and stratum of ctl mice which was inhibited in *Adora1* cKO mice (n = 2 mice in ctl and *Adora1* cKO at 0 hpi, n = 5 mice in ctl and *Adora1* cKO at 6 hpi and 24 hpi). Summary data of **b–d**, **f–h** are presented as the mean ± SEM. Statistical significance in **b, f** was assessed by two-way ANOVA, Fisher's LSD test; statistical significance in (**c, d, g, h**) were assessed by two tailed unpaired Student's t test, ns: p > 0.05. Source data are provided as a Source Data file.

cortex for RNA-seq analysis (Supplementary Fig. 6a). We observed that astrocytes from ctl mice strongly responded to the peripheral LPS challenge in terms of dynamically up and down-regulating genes with time after LPS injection, in line with the previous reports[12]. However, the overall gene expression in A1AR-deficient astrocytes were much less altered upon the LPS injection at 6 and 24 hpi compared to ctl mice (Fig. 4a; Supplementary Fig. 6b; Supplementary Fig. 7). In total, comparing A1AR-deficient astrocytes to ctl astrocytes, we found 12 (4 up, 8 down) differentially expressed genes (DEGs) at 0 hpi (healthy controls), 185 (38 up, 147 down) DEGs at 6 hpi, and 2643 (1465 up, 1178 down) DEGs at 24 hpi (Supplementary Fig. 6c–e). Further Hierarchical Clustering analysis by K-means classified DEGs into 8 clusters (Fig. 4a; Supplementary Fig. 7). At 6 hpi, 125 genes (cluster 1, 5) were upregulated in ctl mice including cytokines and chemokines (e.g., *Ccl2, Ccl5, Ccl9, Cxcl1, Cxcl10*, etc.), transcription factors that promote inflammation (e.g., *Nfkb1a, Nfkb2, Stat3*, etc.), and previously identified marker genes of early responding reactive astrocytes such as interferon-responsive genes (e.g., *Ifit1, 2, 3*) and genes encoding proteins for antigen presentation (e.g., *H2-D1, H2-K1*)[12]. However, in astrocytes of *Adora1* cKO mice the expression of those proinflammatory genes was inhibited. Of note, at 6 hpi several immediate early genes (e.g., *Fos, Jun, and Junb*; cluster 5) were also reactively expressed in ctl astrocytes as previously reported[6,12] and were inhibited in *Adora1* cKO astrocytes, in agreement with the c-Fos immunohistochemistry results. Unlike genes in cluster 5 (e.g., *Fos, Jun, Junb*) that were rapidly down-regulated to baseline level at 24 hpi, on average genes in cluster 1 (e.g., *Ccl5, Ccl9, Cxcl10, Stat3*) were still expressed at high levels in ctl mice but inhibited in A1AR-deficient astrocytes at 24 hpi. In clusters 2, 4, and 6, 918 genes were slightly dysregulated in both ctl and *Adora1* cKO astrocytes at 6 hpi but were either up- (cluster 6) or downregulated (cluster 2 and 4) in ctl astrocytes at 24 hpi.

However, the expression of these genes in *Adora1* cKO astrocytes appeared to be slightly altered at 24 hpi. By Metascape pathway analysis we found that genes in these clusters are responsible for 'DNA repair' (e.g., *Fancm, Msh3, Xpa*, etc. in cluster 6), 'neuronal system' (e.g., *Gabrd, Mpdz*, etc. in cluster 2), and 'cell projection assembly' (e.g., *Fnbp11, Tppp, Nek1*, etc. in cluster 4). We also observed that many genes defined in previous studies as 'A1 astrocyte' (e.g., *H2-D1, H2-T23, Gbp2, Iigp1, Psm8*, etc.) and 'A2 astrocyte' (e.g., *Clcf1, Tgm1, Ptx3*, etc.) markers were decreased in *Adora1* cKO astrocytes at 24 hpi (Supplementary Fig. 6f). Therefore, we assume that this difference in *Adora1* cKO mice at 24 hpi could be regarded as the consequence of reduced astrocytic inflammatory response at 6 hpi which may ameliorate the global neuroinflammation afterwards (see below). For example, the GSEA-KEGG pathway analysis indicated that genes involved in astrocytic inflammatory response pathways (e.g., 'NF-kappa B signaling pathway', 'JAK-STAT signaling pathway', 'IL-17 signaling pathway, NOD-like receptor signaling pathway', 'TNF signaling pathway', etc.)[39,40] were supressed in *Adora1* cKO mice at 6 hpi (Fig. 4b, c). Similarly, GSEA-GO term (biological process, BP) analysis also demonstrated that genes annotated to inflammation-related GO terms (e.g., 'interleukin-1 production', 'toll-like receptor signaling pathway', 'receptor signaling via JAK-STAT', 'I-kappaB/NF-kappaB signaling', 'inflammatory response, cytokine response', 'interferon-gamma production', etc.) were suppressed in A1AR-deficient astrocytes at 6 hpi (Fig. 4d, e). Furthermore, Metascape analysis for upstream transcriptional regulators revealed that the suppressed genes in A1AR-deficient astrocytes at 6 hpi were regulated by transcription factors (e.g., *Jun, Nfkb1, Cebpb, Stat3, Fos*, etc.) known to regulate downstream proinflammatory pathways of reactive astrocytes (Fig. 4f)[39,40]. Taken together, our results demonstrated that in the early phase after a peripheral LPS challenge adenosine triggers via A1ARs the inflammatory response of reactive

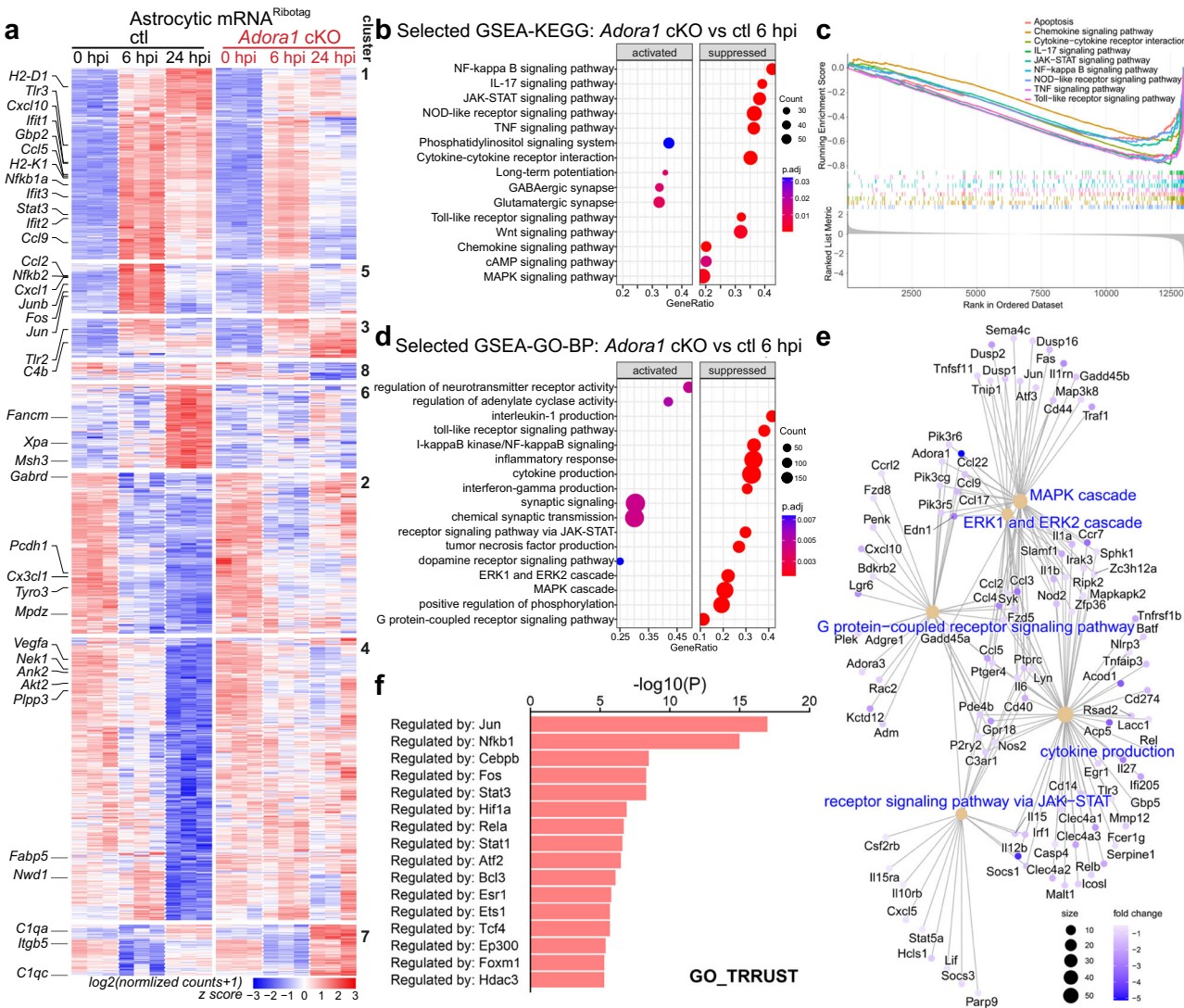

**Fig. 4 | A1AR activation contributes to the altered transcriptional profile of astrocytes upon peripheral LPS challenge. a** Heatmap of altered gene expression (Padj <0.05) from astrocytic mRNA^RiboTag of male *Adora1* cKO and ctl mice in any of the three time points (n = 3 mice per group). Clustering was done with 8 K-means. **b** Selected GSEA-KEGG pathway analysis of the astrocytic RNA-seq dataset between the *Adora1* cKO and ctl groups at 6 hpi. **c** Selected GSEA plot of the enriched KEGG pathways related to inflammation between the *Adora1* cKO and ctl groups at 6 hpi. **d** Selected GSEA-GO pathway analysis of the astrocytic RNA-seq dataset between

the *Adora1* cKO and ctl groups at 6 hpi. **e** Category net plot of selected enriched GO pathways relative to G protein-coupled receptor signaling pathway. The color gradient indicates the fold changes between the *Adora1* cKO and ctl groups. **f** Prediction of transcription regulators following expression pattern of sub-clusters (cluster 1 and 5) by Metascape analysis[77,78]. Statistical significance in **a** was assessed by wald test using Benjamini and Hochberg method, in **b**, **d**, **f** was assessed by hypergeometric test with Benjamini and Hochberg method. Source data are provided as a Source Data file.

---

astrocytes, which may influence the subsequent progression of neuropathology.

**Astrocyte reactivity in the early phase of systemic inflammation boosts global neuroinflammation and microglial reaction**

We next examined the impact of astrocytic A1AR signaling on the global inflammatory response of the brain following a systemic inflammation challenge. We measured 111 different cytokines in the cortex of ctl and *Adora1* cKO mice using a proteomic profiling assay and observed that most of the inflammation-related proteins in the cortex of ctl mice were upregulated after LPS injection at 6 and 24 hpi. However, in *Adora1* cKO mice the levels of several astrocyte specific/related proinflammatory proteins (e.g., CXCL1, CXCL10, ICAM-1, lipocalin-2 (LCN2), MMP3)[40] and many other cytokines or chemokines of multiple origins including microglia and astrocytes (e.g., CCL2, CCL5, IL-1α, CXCL2, etc.)[41] were largely reduced after LPS injection (Fig. 5a, b; Supplementary Fig. 9b, c), in line with the astrocytic RNA^RiboTag

sequencing results. In addition, the differentially expressed protein, LCN2 (Supplementary Fig. 9a), was chosen for Western blot to further confirm the results from the cytokine array. A similarly reduced inflammatory response pattern was detected in the striatum of *Adora1* cKO mice using a small-size (40 cytokines) cytokine array immunodot-blot assay (Fig. 5c, d). These results demonstrate reactive astrocytes upon A1AR activation in the early phase of systemic inflammation exacerbates global neuroinflammation afterwards.

Microglia and astrocytes are key players in neuroinflammation. Previous studies suggested that astrocyte and microglia interact with each other to modulate neuroinflammation[39,42]. Particularly, in the peripheral LPS challenge model, astrocytes can become neurotoxic when triggered by cytokines released by activated microglia[13]. However, these pioneer studies focused on the late phase (after 24 hpi) of the LPS model. Our current results show an adenosine-mediated early response (2–6 hpi) of astrocytes to the LPS challenge that regulate the expression of inflammation-related genes, many of which (e.g., *Ccl2,*

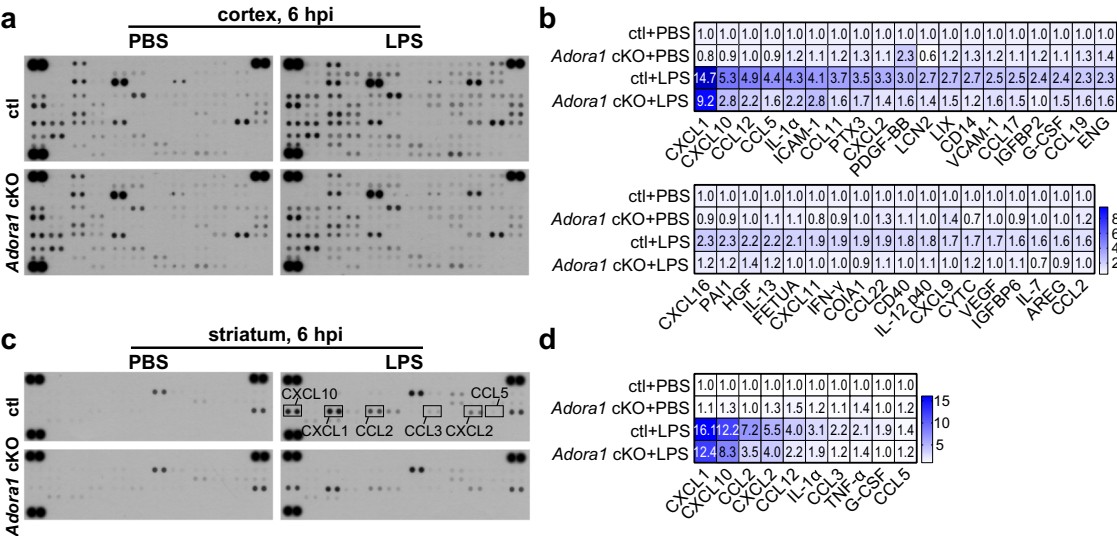

**Fig. 5 | Astrocytic A1AR deficiency reduces global neuroinflammation induced by an LPS challenge. a** Expression of 111 cytokines in the cortex of ctl and *Adora1* cKO mice was measured by a proteomic profiling assay at 6 h after PBS or LPS i.p. injection (samples from 3 mice were pooled for each group). **b** Cytokines with significant changes compared to ctl (PBS) group were shown in the heatmap. Color bar range is between 0.5 and 9.5, out range value was labelled with dark blue.

**c** Expression of 40 cytokines in the striatum of ctl and *Adora1* cKO mice was measured by a proteomic profiling assay at 6 h after PBS or LPS i.p. injection (samples from 3 mice were pooled for each group). **d** Cytokines with significant changes compared to ctl (PBS) group were shown in the heatmap. Color bar range is between 1 and 16.1. Source data are provided as a Source Data file.

*Ccl5, Cxcl1 and Cxcl10*) have been shown to trigger the formation of reactive microglia[40]. Therefore, we hypothesized that astrocytes could modulate microglia activation in the initial phase of systemic inflammation rather than just be the effector of reactive microglia. It is known that reactive microglia produce many inflammatory cytokines via the NF-κB signaling pathway[41]. Therefore, we first examined the reactivity of microglia in the early phase of the LPS model and detected nuclear translocation of p65 (a co-factor of NF-κB) as an indicator of reactive microglia (Fig. 6a).

We found that peripheral LPS challenge increased the proportion of nuclear p65⁺ microglia in the cortex of ctl mice from ~13% at 2 hpi to ~90% at 6 hpi and ~30% at 24 hpi, whereas such proportion in *Adora1* cKO mice was reduced to ~9%, ~55% and ~17% at 2, 6 and 24 hpi, respectively (Fig. 6a, b). Similarly, in the hippocampus and striatum the nuclear p65⁺ microglia were also reduced proportionally in the *Adora1* cKO mice at 6 hpi (Fig. 6c, d). Therefore, our results revealed that a systemic LPS challenge initiates astrocyte activation in the early phase via A1AR signaling to augment reactive microglial inflammatory response. In addition, systemic inflammation challenges can enhance the phagocytosis of microglia which contributes to the depression-like behaviour. Therefore, we quantified the CD68⁺ lysosome volume in microglia to assess the phagocytic ability of microglia in our model (Fig. 6e). We found that the volume of CD68⁺ lysosome (in proportion to microglial volume) in the cortex of ctl mice increased from ~1% at 0 hpi to ~4% at 6 hpi and further increased to ~11% at 24 hpi. However, in *Adora1* cKO mice such volume enlargement was significantly inhibited to ~2% and ~5% at 6 and 24 hpi, respectively, indicating the microglial phagocytosis was inhibited by the astrocytic A1AR deficiency (Fig. 6e, f). To further confirm these ameliorated phenotypic alterations of microglia, we analysed microglial morphologies in *Adora1* cKO and ctl mice (Fig. 6g) by IMARIS software. In the healthy control group (0 hpi), microglia of *Adora1* cKO and ctl mice displayed comparable morphologies. However, 6 or 24 hours after LPS injection microglia in *Adora1* cKO mice showed more intersections in the Sholl-analysis, more total process length, and a larger occupied area (Fig. 6g–i), indicating reduced microglial activation in the absence of astrocytic A1AR signaling. In line with the morphological changes, we detected that the gene expression levels of microglial homeostasis markers such

as *P2ry12* were reduced at 6 hpi in both groups. However, at 24 hpi the *P2ry12* expression in *Adora1* cKO mice recovered to the healthy level while in ctl mice it was still significantly reduced (Fig. 6j). Taken together, our results demonstrate that A1AR signaling triggers early reactive astrocytes to provoke the pathological phenotype formation of reactive microglia after a systemic inflammation challenge.

## Astrocytic A1AR signaling contributes to impaired BBB function and peripheral immune cell infiltration in systemic inflammation

Next, we investigated BBB function-related parameters in *Adora1* cKO and ctl mice following a peripheral LPS challenge. Systemic inflammation induces the CCL5-CCR5 axis-dependent migration of microglia to blood vessels, which further impairs the BBB[18]. We found that after LPS injection the perivascular microglia in ctl mice increased proportionally from ~35% at 0 hpi to ~50% at 6 and 24 hpi, while the proportion of perivascular microglia in *Adora1* cKO mice was only increased from ~35% to ~43% at 6 and ~45% at 24 hpi, indicating a significant reduction (~50%) of newly recruited perivascular microglia in *Adora1* cKO mice after peripheral LPS injection (Supplementary Fig. 8a–c), in line with the reduction of CCL5 expression in the cKO mice. To assess the BBB integrity, we injected EB i.v. to mice immediately after the LPS injection and allowed EB to circulate for 24 hours. Afterwards, we measured EB extravasation in brain tissue. We detected less EB in the brain of *Adora1* cKO mice than in ctl (~10 vs. 22 μg/g tissue, Supplementary Fig. 8d), indicating less disruption of the BBB in *Adora1* cKO mice. Systemic inflammation promotes infiltration of peripheral immune cells into the brain parenchyma, attributable to the disrupted BBB as well as an upregulated expression of chemokines (e.g., CXCL1)[43] and extracellular matrix proteins (e.g., ICAM-1)[44]. Therefore, we used flow cytometry to identify different immune cell populations in the brain (Supplementary Fig. 8e). We found that at 6 hpi, the proportions of infiltrated monocytes (CD11b⁺/CD45$^{low+}$/Ly6C$^{neg-}$/Ly6G$^{high+}$; ctl: ~4% vs *Adora1* cKO: ~2%), neutrophils (CD11b⁺/CD45$^{low+}$/Ly6C$^{inter+}$/Ly6G$^{high+}$; ctl:~4% vs *Adora1* cKO: ~2%), and T cells (CD45⁺/CD3$^{high+}$/Ly6G$^{neg-}$; ctl: ~4% vs *Adora1* cKO: ~2%) were significantly reduced in *Adora1* cKO mice among all immune cells (Supplementary Fig. 8f, j). Furthermore,

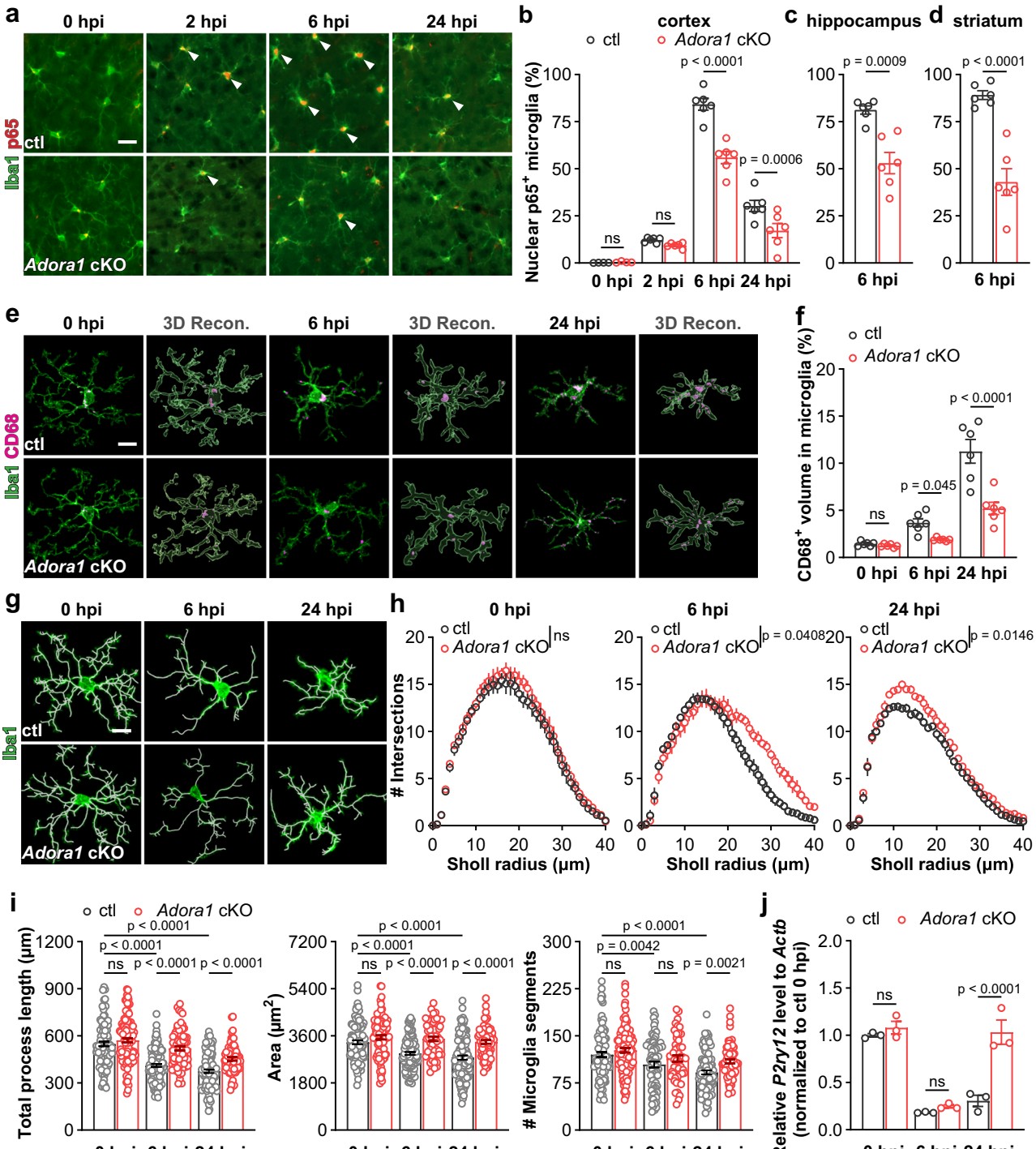

**Fig. 6 | Astrocytic A1AR deficiency inhibits microglial reaction upon LPS challenge. a** Representative images of p65 immunoreactivity in Iba1⁺ microglia post LPS injection. Arrowheads indicated Iba1⁺ microglia with nuclear P65. Scale bar = 20 μm. **b**–**d** Proportions of nuclear p65⁺ microglia in the cortex, hippocampus, and striatum of *Adora1* cKO mice were reduced post LPS injection compared to ctl mice. (n = 4 mice in ctl and *Adora1* cKO at 0 hpi, n = 6 mice in ctl and *Adora1* cKO at 2 hpi, 6 hpi, and 24 hpi). **e** Representative images and 3D reconstructions of CD68⁺ and Iba1⁺ volume post the LPS injection by IMARIS. Scale bar = 10 μm. **f** Percentage of CD68⁺ volume in microglia in the cortex of *Adora1* cKO mice were reduced post LPS injection compared to ctl mice (n = 6 in ctl and *Adora1* cKO at 0 hpi, 6 hpi, and 24 hpi). **g** Morphology and 3D reconstruction of Iba1⁺

microglia post the LPS injection. Scale bar = 10 μm. **h** Sholl analysis of Iba1⁺ microglia at 0 hpi, 6 hpi, 24 hpi (n = 3 mice per group). **i** Total process length, area, and segments number of Iba1⁺ microglia in *Adora1* cKO and ctl mice post LPS injection obtained from the IMARIS-based morphological analysis (n = 3 mice per group). **j** Relative expression of *P2ry12* was elevated in the cortex of *Adora1* cKO mice at 24 hpi (n = 3 mice per group). Summary data are presented as the mean ± SEM. Statistical significance in **b**, **f**, **h**, **i**, **j** were assessed using a two-way ANOVA, Fisher's LSD test. Statistical significance in **c**, **d** is assessed using two tailed unpaired Student's t test; ns: p > 0.05. Source data are provided as a Source Data file.

we performed Ly6B immunostaining to detect neutrophils at 24 hpi. We found that the entry of Ly6B[+] cells into the brain parenchyma was largely reduced 24 hours after the peripheral LPS injection in cKO mice compared to ctl (Supplementary Fig. 8h, i), in line with our current finding of reduced CXCL1 and ICAM-1 expression in *Adora1* cKO mice. Taken together, upon a systemic inflammation challenge astrocytic A1AR deficiency inhibited microglial reaction and neuroinflammation which ameliorated the impairment of the BBB and reduced peripheral immune cell infiltration.

**A1AR-mediated astrocyte activation promotes aberrant neuronal functions and behavioural deficits in systemic inflammation**
Systemic inflammation is also known to alter neuronal hyperactivity. For example, two hours after peripheral administration of LPS, pericytes are triggered to release CCL2, which subsequently determines neuronal hyperactivity[5]. Therefore, we performed c-Fos and NeuN (a neuron marker) double immunostaining and studied neuronal activation after a peripheral LPS challenge. In contrast to astrocytes, we observed that c-Fos expression in cortical neurons in ctl mice gradually increased from 6 hpi to 24 hpi. Although c-Fos expression was still elevated in neurons from *Adora1* cKO mice compared with basal levels (0 hpi), it was lower than in ctl mice at 6 and 24 hpi (Fig. 7a, b). Similarly, we detected lower c-Fos levels in hippocampal and in striatal NeuN[+] neurons of *Adora1* cKO mice at 24 hpi (Fig. 7c, d). Thereby, these results further confirm that astrocytic A1ARs contribute to neuronal activation upon the challenge of systemic inflammation.

Neuroinflammation induced by systemic endotoxin has been shown to affect neuronal network plasticity in terms of impaired long-term potentiation (LTP)[45–47]. To study the impact of astrocyte-specific A1AR deficiency on neuronal plasticity following a peripheral LPS challenge, we performed extracellular electrophysiological recordings on acute slices prepared from dorsal hippocampi of *Adora1* cKO and ctl mice. Neurons from both groups displayed comparable LTP after PBS injection. However, LTP was impaired after LPS injection in ctl mice at 6 and 24 hpi, whereas it was preserved in cKO mice (Fig. 7e–g). These results indicate that adenosine triggers an astrocytic inflammatory response as a detrimental regulator of neuronal functions after systemic LPS challenge.

Neuroinflammation induced by peripheral LPS administration leads to fatigue, muscle weakness, and other sickness behaviour in mice (e.g., hunched posture, reduced food and water intake, etc.) that peaks between 2 and 6 hpi and gradually wanes. Subsequently, the depression-like behaviour (e.g., reduced locomotion in the open-field test, loss of preference to drink sweetened water, etc.) gradually develops to reach the peak 24 hours later[3,14]. Therefore, we performed the open-field test to assess the general exploratory locomotion of mice as a total readout of behavioural deficits at 24 hpi (Fig. 7h). We observed ctl and *Adora1* cKO mice injected with PBS performed similarly in the open-field test. As expected, after LPS injection all ctl and *Adora1* cKO mice drastically reduced locomotion, whereas *Adora1* cKO mice moved longer distances than ctl mice (Fig. 7i, j), suggesting that the deficiency of astrocytic A1ARs could alleviate behavioural deficits induced by systemic inflammation. However, the time spent in the central area, a parameter often used to assess depression/anxiety of mice, was heavily reduced in both groups, though the *Adora1* cKO mice showed a tendency of increasing the time. Concerning other aspects of the sickness such as fatigue may influence the interpretation of the time spend in the central area, we used the sucrose-preference test to further specifically evaluate anhedonia, another hallmark of systemic inflammation-induced depression-like behaviour (Fig. 7h). We observed that peripheral LPS challenge could largely abolish the sucrose preference of ctl mice from 24-48 hpi (~45%) which partially recovered to ~60% from 48–72 hpi. However, the sucrose preference of *Adora1* cKO mice was only partially impaired to ~65% from 24-48 hpi, and almost fully recovered to the healthy level from 48–72 hpi (~80%),

again indicating activation of astrocytic A1ARs contributes to behavioural deficits induced by systemic inflammation (Fig. 7k).

**Enhancing Gi signaling in A1AR-deficient astrocytes restored neuroinflammation upon peripheral LPS challenge**
A1ARs are $G_{i/o}$ protein-coupled receptors. Therefore, a weakened Gi signaling can be assumed in astrocytes of cKO mice. The Gi signaling could be artificially rescued using Designer Receptors Exclusively Activated by Designer Drugs (*DREADD*)-based chemogenetic tool hM4Di upon administration of its specific agonist CNO (clozapine N-oxide). Since previous studies highlighted that reactive microglia in the mouse striatum is a major contributor to depression-like behaviours after peripheral LPS challenge[48], we stereotactically injected AAV-GFAP-hM4Di-mCherry or AAV-GFAP-tdTomato into the striatum of *Adora1* cKO and ctl mice to specifically express hM4Di (indicated by mCherry) or tdTomato (tdT, as control) in astrocytes. To study whether A1AR-mediated astrocyte activation at the early phase of systemic inflammation is crucial to provoke the subsequent neuroinflammation, we injected CNO (2 mg/kg, half-life = ~0.5 h) to AAV-delivered mice to evoke Gi signaling at 2 and 4 h post peripheral LPS challenge (Fig. 8a–c). At 6 hpi, we observed that in the *Adora1* cKO mice c-Fos expression was more enhanced in hM4Di-expressing rather than in tdT-expressing astrocytes, indicating that Gi signaling could further activate A1AR-deficient astrocytes post LPS injection (Fig. 8d, e). Concomitantly, we found increased nuclear p65[+] microglia in the regions with hM4Di-expressing astrocytes compared to tdT-expressing astrocytes in *Adora1* cKO mice, suggesting that enhancing astrocytic Gi signaling could promote reactive microglial inflammatory response at the early stage of systemic inflammation (Fig. 8f, g). To assess the long-term effect of activation of Gi signaling in A1AR-deficient astrocytes, we measured the expression of proinflammatory factors in AAV-infected brain regions by cytokine array 24 h post LPS injection. We observed that the expression of the factors (e.g., CCL2, CCL5, CCL3, CXCL1, CXCL2, CXCL10, G-CSF, etc.) inhibited in the tdT-expressing *Adora1* cKO mice (compared to tdT-expressing ctl mice) could be enhanced to certain extents in hM4Di-expressing *Adora1* cKO mice (Fig. 8h, i). In addition, a similar effect could be detected in the cortex of *Adora1* cKO mice with astrocytic hM4Di activation (Supplementary Fig. 9d, e), suggesting a consistent effect of Gi signaling in astrocytes from different brain regions to promote neuroinflammation. These results were in line with our previous pharmacological intervention experiments showing the A1AR antagonist DPCPX treatment could reduce neuroinflammation during the early phase of systemic inflammation (Supplementary Fig. 2d, e). Next, we performed an open-field test and found that the alleviated behavioural deficit in *Adora1* cKO mice upon LPS challenge could be exacerbated to the level of ctl mice upon astrocytic Gi signaling activation (Fig. 8j–l). Likewise, we observed ctl mice treated with DPCPX at 2 and 4 hpi showed increased locomotion in the open-field test at 24 hpi, suggesting an ameliorated behavioural deficits (Supplementary Fig. 10a–c). In summary, our results strongly suggest that early astrocyte reaction to A1AR signaling in systemic inflammation plays an important role in promoting neuroinflammation to drive pathogenesis of SAE and targeting A1ARs can be a therapeutic approach for SAE.

## Discussion
Adenosine has been widely recognized as a guardian molecule that executes protective function in and outside of the CNS. For example, in the periphery adenosine reduced inflammation by triggering A2a, A2b, and A3 ARs of immune cells (e.g., macrophages, lymphocytes, neutrophils, etc.) as demonstrated, among others, in autoimmune diseases (e.g., rheumatoid arthritis), inflammatory bone loss, intestinal inflammation[16]. In the CNS, adenosine is known to trigger pre- and post-synaptic A1ARs to reduce neuronal activity and, thereby, to alleviate epilepsy and pain[16]. However, recent studies also suggest that the

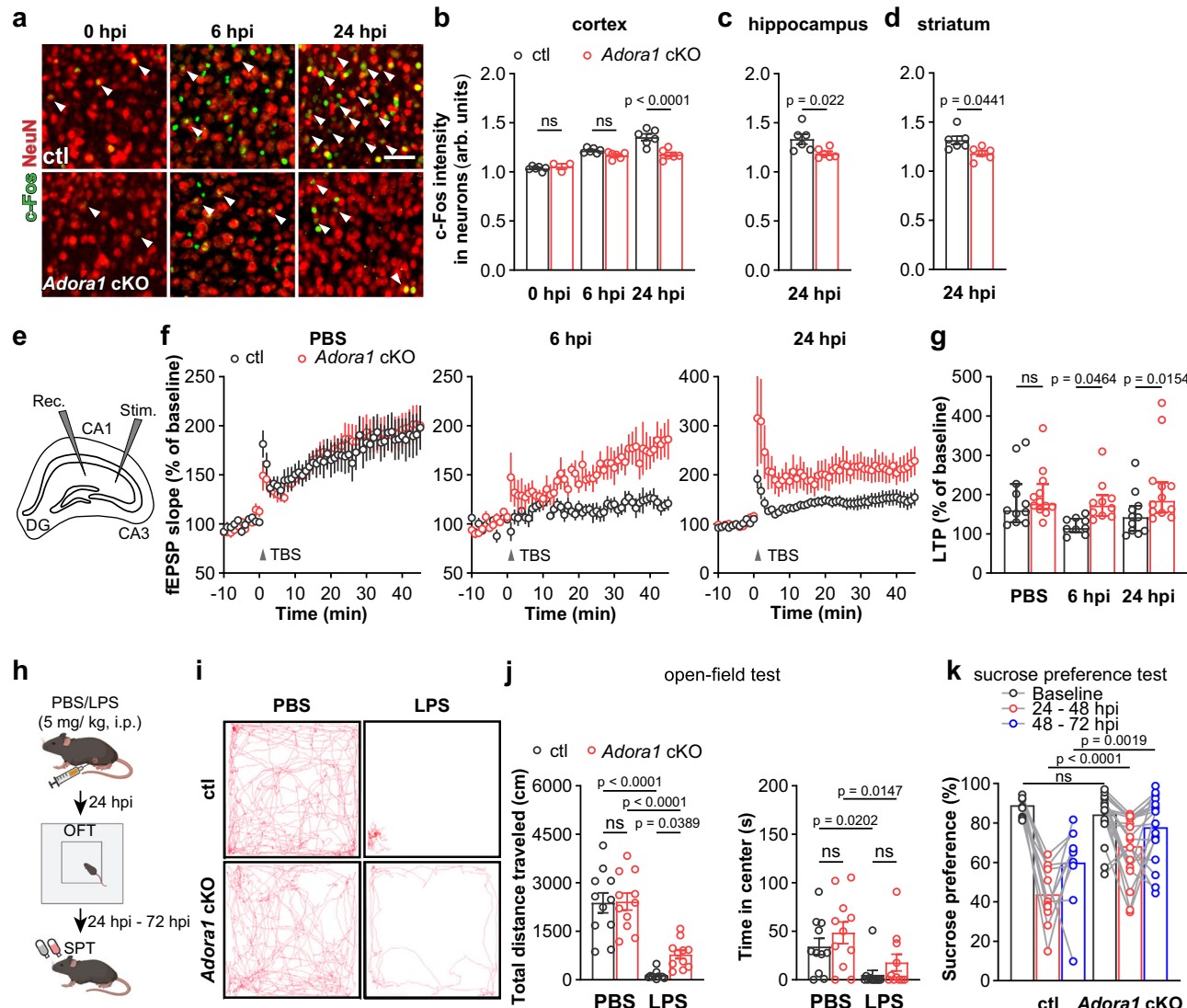

**Fig. 7 | Astrocytic A1AR deficiency prevents neuronal dysfunction and ameliorates depression-like behaviour of the mice after LPS treatment.**
**a** Representative images of c-Fos immunoreactivity in cortical NeuN⁺ neurons after LPS injection. Scale bar = 20 μm. **b–d** c-Fos immunofluorescence intensities (arb. units) in NeuN⁺ neurons in the cortex, hippocampus, and striatum of *Adora1* cKO mice were reduced compared to ctl mice at 6 and 24 hpi (n = 6 mice in ctl and *Adora1* cKO at 0 hpi, n = 6 mice in ctl and *Adora1* cKO at 6 hpi, 24 hpi). **e** Graphical description of LTP measurement protocol. **f** Scatter plots showing LTP induced by stimulation of Schaffer collateral (SC) − cornu ammonis (CA) 1 synapses with TBS in acute hippocampal slices from ctl and *Adora1* cKO at 0 hpi, 6 hpi, 24 hpi. Averaged fEPSP are plotted versus time (n = 11 slices in ctl at 0 hpi, n = 12 slices in *Adora1* cKO at 0 hpi, n = 9 slices in ctl at 6 hpi, n = 10 slices in *Adora1* cKO at 6 hpi, n = 11 slices in ctl at 24 hpi, n = 12 slices in *Adora1* cKO at 24 hpi). **g** LTP evoked in hippocampi of *Adora1* cKO and ctl mice. (n = 11 slices in ctl at 0 hpi, n = 12 slices in Adora1 cKO at 0

hpi, n = 9 slices in ctl at 6 hpi, n = 10 slices in Adora1 cKO at 6 hpi, n = 11 slices in ctl at 24 hpi, n = 12 slices in Adora1 cKO at 24 hpi). **h** Schematic illustration of open-field test and sucrose preference test post PBS/LPS injection. **i** Representative trajectory analysis of ctl and *Adora1* cKO mice in 10 min in the open-field test at 24 hpi. **j** *Adora1* cKO mice displayed protected locomotion compared to ctl mice at 24 hpi (n = 11 mice in each group). **k** *Adora1* cKO mice displayed less LPS-induced decrease in sucrose preference than ctl after LPS injection (n = 11 mice in ctl group, n = 20 mice in *Adora1* cKO group). Summary data are presented as the mean ± SEM in **b**–**d**, **j**, **k** and median ± IQR in **g**. Statistical significance in **b**, **g**, **j**, **k** were assessed using a two-way ANOVA, Fisher's LSD test; statistical significance in **c**, **d** was assessed using a two tailed unpaired Student's t test. ns: p > 0.05,. Source data are provided as a Source Data file. Panel **h** was created with BioRender.com released under a Creative Commons Attribution-NonCommercial-NoDerivs 4.0 International license.

overproduction of adenosine could enhance neuroinflammation, for example, via microglial A2a AR in Parkinson disease (PD)[15,17]. In the current study, we revealed that peripheral administration of adenosine or its analogues can directly trigger inflammatory responses in the brain mediated at least partially by astrocytic A1ARs. Using peripheral LPS challenge, we provide further evidence that systemic inflammation-induced adenosine rise in the brain plays an important role via astrocytic A1AR signaling in exacerbating neuroinflammation to drive the pathogenesis of the SAE. Therefore, like its precursor ATP, adenosine can also behave like a DAMP (damage-associated molecular patterns) molecule for neuroinflammation, possibly in a context-dependent manner.

Adenosine is mainly formed from the sequential ATP metabolism by means of a series of hydrolases, among which the ecto-5′-nucleotidase Nt5e (also known as CD73) performs the last step converting AMP to adenosine[49,50]. Under cellular stress conditions, such as inflammation or hypoxia, extracellular ATP/adenosine can drastically increase[51,52]. Recent studies using a novel genetically modified ATP sensor (GRAB_{ATPL0}) provided direct evidence that extracellular ATP levels in the mouse brain were increased after peripheral LPS challenge[11]. Therefore, it is conceivable that increased extracellular ATP in the brain may contribute to adenosine elevation as well. Indeed, here we detected elevated extracellular adenosine in the brain

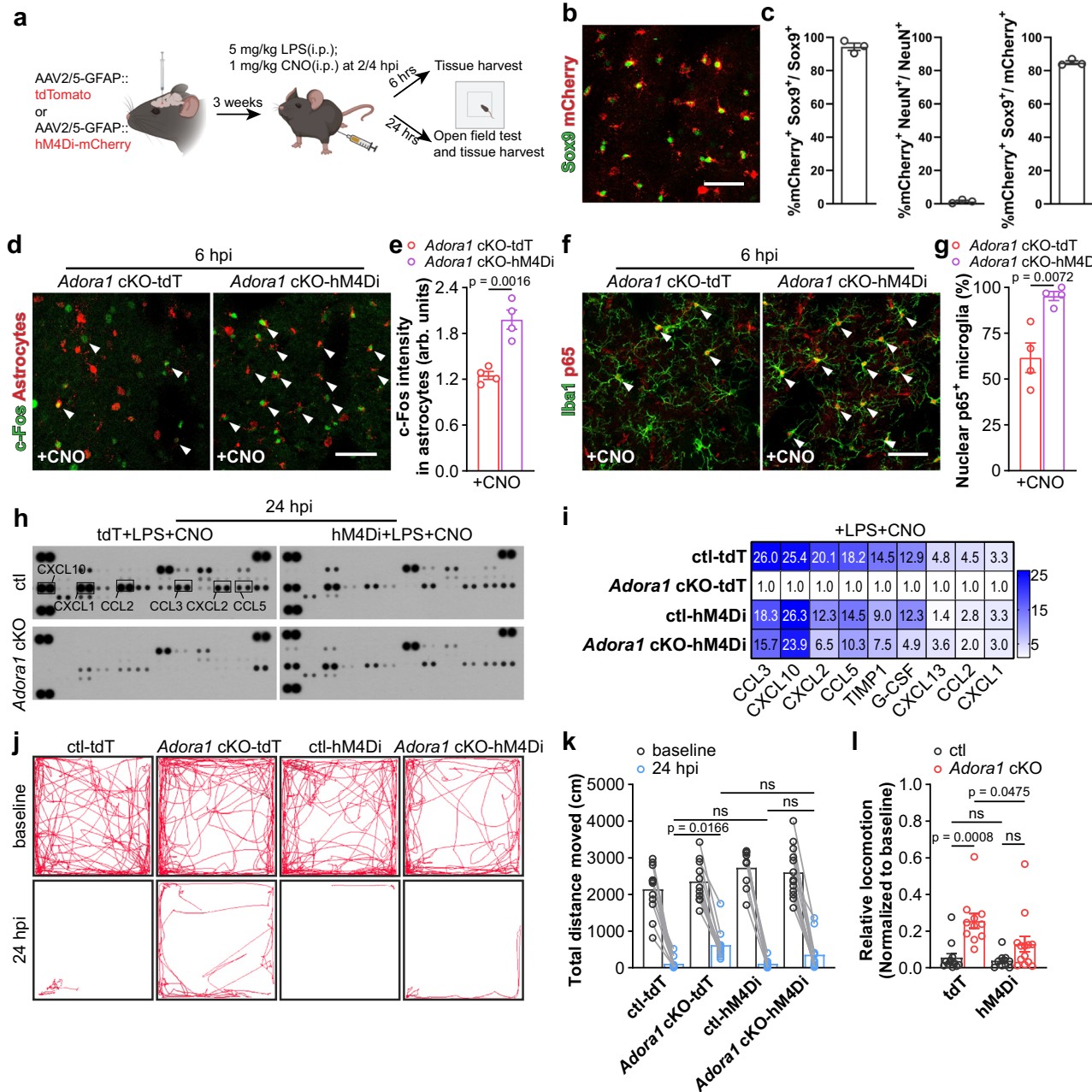

**Fig. 8 | Activation of Gi signaling in A1AR deficit astrocytes restores the inflammatory response to peripheral LPS challenge. a** Schematic illustration of activation of Gi signaling in A1AR deficit astrocytes experiment employing DREADD hM4Di and experimental plan. **b** Representative image of hM4Di expression indicated by mCherry in the Sox9⁺ astrocytes. Scale bar = 50 μm. **c** hM4Di was expressed in 94% of striatal astrocytes, with >84% specificity (n = 3 mice). **d** Representative images of immunolabeled reactive A1AR-deficient astrocytes by c-Fos immunostaining after LPS and CNO injection. Scale bar = 50 μm. **e** Activation of hM4Di increased c-Fos expression in hM4Di-mCherry⁺ A1AR-deficient astrocytes after LPS injection (n = 4 mice per group). **f** Representative images of immunoreactivity of p65 in Iba1⁺ microglia in *Adora1* cKO mice with astrocytic tdT or hM4Di expression after LPS and CNO injection. Scale bars = 50 μm. Arrowheads indicates Iba1⁺ microglia with nuclear p65 expression. **g** Nuclear p65⁺ microglia was increased in hM4Di-expressing *Adora1* cKO mice after LPS and CNO injection compared to ctl mice (n = 4 mice per group). **h** The

expression of 40 cytokines in the striatum of AAV-infected ctl and *Adora1* cKO mice was measured by a proteomic profiling assay after LPS and CNO injection. **i** Enhancing Gi signaling in *Adora1* cKO mice increased cytokine expression after LPS and CNO injection (samples from 3 mice were pooled for each group). **j** Representative trajectory analysis digitally tracked movement of ctl and *Adora1* cKO mice injected with LPS and CNO in the open-field test at 24 hpi. **k, l** Enhancing Gi signaling in A1-deficit astrocyte reduced locomotion in the open-field test (n = 12 mice in ctl-tdT+CNO group, n = 11 mice in *Adora1* cKO-tdT+CNO group, n = 9 mice in ctl-hM4Di+CNO group, n = 13 mice in *Adora1* cKO-hM4Di+CNO group). Summary data are presented as the mean ± SEM. Statistical significance in **e, g** was assessed using two tailed unpaired Student's t test; statistical significance in **k, l** was assessed using a two-way ANOVA, Fisher's LSD test, ns: p > 0.05. Source data are provided as a Source Data file. Panel **a** was created with BioRender.com released under a Creative Commons Attribution-NonCommercial-NoDerivs 4.0 International license.

parenchyma upon LPS injection employing in vivo 2P-LSM imaging of GRAB_Ado1.0 and this effect lasted at least 24 h post the LPS challenge. In addition, adenosine can be transported by equilibrative nucleoside transporters (ENTs) or can bypass, in virtue of its small size, tight junctions (TJs) of endothelial cells[53,54]. Furthermore, adenosine can increase the permeability of the BBB through A1 and A2a ARs in endothelial cells, which facilitate the infiltration of peripheral immune cells (or even bacteria) expressing CD73 into the brain parenchyma[19,24,55,56]. To determine whether humoral adenosine could not only open the BBB but also further diffuse into the brain parenchyma, we injected adenosine i.p to mice during the 2P-LSM recording of GRAB_Ado1.0 signals. We were able to detect increasing GRAB_Ado1.0 signals in the brain along with the entry of the peripherally administered adenosine into cerebral blood flow indicated by SR101 in a dose-dependent manner. Although locally generated extracellular adenosine was reported to be quickly eliminated within seconds, previous studies have shown that i.p. injected adenosine could remain in the blood for at least one hour[22,24]. Therefore, our 2P-LSM results of adenosine administration strongly suggest that peripherally administered adenosine could either pass through the BBB or trigger endothelial cells to release ATP/adenosine (or both), thereby increasing the extracellular adenosine level in the brain parenchyma. We also demonstrated (as found in septic patients and volunteers receiving LPS injection[8,57]) that the plasma level of adenosine was increased by the LPS challenge gradually peaking at 6 hpi, concomitant with the progression of astrocytic c-Fos and p-STAT3 expression, BBB breakdown and sickness behaviour of the mice. Moreover, considering that the BBB integrity was impaired by systemic inflammation, which could permit even more adenosine entry from circulation to the brain parenchyma, it is conceivable that systemic inflammation-increased humoral adenosine adds to locally generated adenosine to affect cells building the BBB (e.g., astrocytes and pericytes) at least during the early phase of systemic inflammation, provoking neuroinflammation.

In line with recent studies[12], our c-Fos and p-STAT3 immunohistochemistry as well as RNA^RiboTag sequencing results revealed that astrocytes reacted rapidly within hours to a systemic inflammation challenge and underwent dynamic transcriptomic changes. Although many molecules such as TNFα, IL-1α from reactive microglia, ATP, NO, etc. have been identified as triggers of astrocyte reactivity[40], the exact molecules that initiate/boost such fast astrocyte reactivity to systemic inflammation are still undefined[58]. Here, we demonstrated that the removal of A1ARs from astrocytes could largely prevent such fast astrocyte reactivity upon LPS injection. Moreover, adenosine and its analogues were able to elicit a neuroinflammatory response in healthy mice, which was strongly attenuated by lack of astrocytic A1AR. Our results suggest that the formation of reactive astrocytes in the early phase of systemic inflammation requires proinflammatory cytokines from the circulation or other CNS cells, but small molecules from the metabolism could play important roles as well. Here, adenosine is proposed as the first identified small molecule. It triggers the early astrocytic response to systemic inflammation via A1AR signaling.

Astrocytes express a plethora of GPCRs on the plasma membrane to sense the extracellular signal molecules such as glutamate, ATP, and adenosine which induce intracellular Ca^{2+} signals. Previous chemogenetic studies suggested that activation of Gi sigaling in astrocytes leads to Ca^{2+} signal elevation via phospholipase C (PLC) pathway[33,37]. In the current study using brain slice recording in *Adora1* cKO mice we detected more astrocytic spontaneous Ca^{2+} signals (microdomains) with lower amplitude in the presence of TTX, which led to more ROAs with smaller sizes. Although the intracellular mechanisms attributed to such an observation are not yet clear, these results suggest astrocytes respond to endogenous extracellular adenosine via A1ARs under physiological conditions. Nevertheless, on the other hand, astrocytic-A1AR deficiency does not seem to heavily influence the physiological functions of astrocytes in terms of minor alterations of gene

expressions, microglia reactivity, BBB integrity, neuronal plasticity, and general behavioural performance in current tests. However, recent studies revealed that pathological stimuli trigger aberrant Ca^{2+} signaling in astrocytes which promotes their reactivity and increases neuroinflammation[59,60]. Therefore, it is conceivable that after LPS injection, the increased extracellular adenosine acting on A1ARs could evoke aberrant astrocytic Ca^{2+} signals, though in vivo Ca^{2+} live imaging will be required to further investigate this scenario. Furthermore, our mRNA^RiboTag sequencing and immunohistochemical results demonstrated that several signaling pathways (e.g., Fos-Jun, JAK-STAT3, MAPK) are involved in regulating the A1AR-mediated inflammatory response of astrocytes. Given that intracellular Ca^{2+} activities were suggested to be involved in the regulation of these signaling pathways[33,37,61,62], we could hypothesize that at the early phase of systemic inflammation A1AR activation-induced aberrant Ca^{2+} signals of astrocytes enhance diverse intracellular signaling pathways to provoke the inflammation response as suggested by the RNA profiling analyses.

Recent transcriptomic studies demonstrated astrocytes and microglia are both fast responders at the early phase of systemic inflammation[6,12,45]. Although reactive astrocytes were usually considered as the long-term effector of reactive microglia especially in the systemic inflammation model[58], whether this is the case for the early astrocyte reactivity is unclear. Our astrocytic RNA^RiboTag sequencing results revealed that astrocytes at 6 hpi were triggered to release a series of inflammatory factors that are known to activate microglia via NF-κB activation, which were largely inhibited upon the ablation of A1AR signaling. Compensating the downstream Gi signaling by DREADD hM4Di strategy in the A1AR-deficient astrocytes before 6 hpi could largely restore the NF-κB activation of microglia and global neuroinflammation. Therefore, we provide evidence that astrocytes at the early phase of systemic inflammation behave more like a booster of reactive microglia. Furthermore, the immune signal from the inflamed circulation was shown to be relayed by cells building the BBB (i.e., endothelial cells and pericytes) to the brain parenchyma[5–7]. As another major component of the BBB, each astrocyte extends its end-feet to wrap at least one blood vessel[63]. Our results suggest that, in addition to endothelial cells/pericytes, immune signals mediated by adenosine from the circulation stimulate astrocytes to subsequently enhance microglia reaction at the initial phase of the systemic inflammation. Thereby, reactive microglia may in turn modulate astrocyte reactivity in terms of the temporally changed transcriptome towards the neurotoxic A1-type in the late phase of systemic inflammation (24 hpi). However, such adenosine-regulated reciprocal interactions between astrocytes and microglia needs further investigation in other neuroinflammatory diseases such as multiple sclerosis (MS). In addition, future studies need to determine the dynamics of the interactome between astrocytes and microglia (and other inflammation-related cells such oligodendrocyte precursor cells) to gain insights into the progression of neuroinflammation in sepsis.

In conclusion, we provide evidence that adenosine, a ubiquitous signaling molecule, behaves as an immune mediator of systemic inflammation to exacerbate neuroinflammation via astrocytic A1ARs, thereby driving the pathogenesis of SAE. Since A1ARs are the most abundant ARs expressed in the brain, understanding how A1ARs, which are expressed in other cell types (i.e., pericytes, microglia, oligodendrocyte precursor cells) as well, contribute to neuroinflammation will enhance our knowledge on adenosine signalling for inflammation-related CNS diseases and might be the basis for future therapeutical approaches.

## Methods
### Animals
This study involved the following animals maintained on C57BL/6 background: A1AR^{fl/fl} mice[28]; Glast-CreERT2(CT2) mice[64], Cx3cr1-CreERT2 mice[65], NG2-CreERT2 mice[66]; RiboTag mice (Rlp22HA)[29];

GCaMP3 reporter mice (Rosa 26-CAG-lsl-GCAMP3)[30]. To obtain cell type-specific A1AR knockout mice, A1AR$^{fl/fl}$ mice were crossed to Glast-CreERT2 mice, Cx3cr1-CreERT2 mice, and NG2-CreERT2 mice to induce the specific knockout of A1ARs in astrocytes, microglia, and OPCs/pericytes, respectively, upon tamoxifen administration. For Fig. 1, Fig. 2b, c, and Supplementary Fig. 1, C57BL/6, A1AR$^{wt/wt}$xGlast$^{ct2/wt}$ and A1AR$^{fl/fl}$xGlast$^{wt/wt}$ littermate mice were used as 'wild-type' (wt) control (ctl) mice without tamoxifen injection. For other experiments, A1AR$^{wt/wt}$xGlast$^{ct2/wt}$ and A1AR$^{fl/fl}$xGlast$^{wt/wt}$ littermate mice were used as control (ctl) mice, also receiving tamoxifen injection.

RiboTag mice (Rlp22HA) were introduced to immunoprecipitate ribosome-associated translated mRNA in astrocytes upon breeding with A1AR$^{fl/fl}$xGlast-CreERT2 mice. To visualize Ca$^{2+}$ activity in astrocyte, GCaMP3 reporter mice were introduced to A1AR$^{fl/fl}$xGlast-CreERT2 mice.

Mice of both sexes with 11–13 weeks age were used for most experiments (details in source data). All animals were maintained at the animal facility of the CIPMM under temperature- (22 °C ± 2 °C) and humidity- (45–65%) controlled conditions with 12 h light/dark cycle with an *ad libitum* supply of food and water. Animal husbandry and procedures were performed at the animal facility of CIPMM, University of Saarland according to European and German guidelines for the welfare of experimental animals. Animal experiments were approved by the Saarland state's "Landesamt für Gesundheit und Ver-braucherschutz" in Saarbrücken/Germany (animal license number: 65/2013, 12/2014, 34/2016, 36/2016, 03/2021 and 08/2021).

### Tamoxifen (TAM), LPS treatment
As previously described[67], tamoxifen (CC99648, Carbolution) was dissolved in Miglyol (3274, Caesar & Loretz) at a concentration of 10 mg/ml and injected intraperitoneally to all mice crossed with CreERT2-driver mouse lines (100 mg/kg per body weight) for five consecutive days at the age of 4 weeks.

For the endotoxin challenge, 13 weeks old mice were injected intraperitoneally (i.p.) with 5 mg/kg of LPS (L2880, *E. coli* O55:B5, Sigma) or endotoxin-free PBS between 1 p.m. to 3 p.m. (CET) and kept for a maximum of 72 h post injection[12].

### Anesthesia for the post-mortem animal experiments
All mice were anesthetized by i.p. injection of a low dose of ketamine/xylazin (120 and 12 mg/kg of body weight, respectively), followed by a lethal dose of ketamine/xylazin (240 and 24 mg/kg of body weight, respectively).

### Plasma adenosine measurement
To measure plasma adenosine level, 0.4 ml blood was collected from the right ventricle of deeply anesthetized mice with a syringe containing 0.4 ml ice-cold stop solution (220 μM APCP (3633, Tocris), 100 μM NBMPR (N2255, Sigma), 40 μM dipyridamole (D9766, Sigma), 13.2 mM Na$_2$EDTA, 118 mM NaCl, 5 mM KCl), to block adenosine formation and transport. After centrifugation (1000 × *g*, 4 °C, 10 min), plasma was stored at −80 °C until analysis.

The plasma adenosine concentration was quantified with an adenosine assay kit (ab211094, Abcam) according to the protocol. The fluorescence was measured at excitation/emission = 535/587 nm.

### Evaluation of blood brain barrier (BBB) permeability by Evans blue (EB)
For Fig. 1d EB (4% wt/vol in saline, 2 mL/kg) was administered through the tail vein 2 h before mice were euthanized. For Fig. 1c, mice were injected i.p. EB (4% wt/vol in saline, 10 mL/kg) 10 min prior to the i.p. injection of adenosine and were euthanized 2 h post the adenosine injection. Afterwards, Mice were perfused through the left ventricle with 10 ml ice-cold PBS to remove intravascular albumin-EB. Then half of the brains were quickly removed, weighed, and homogenized in

1 mL of PBS. After the first homogenization, 0.5 mL homogenization were mixed with 1.5 mL of 50% trichloroacetic acid, followed by vortex mixing for 2 min. After centrifugation (10,000 × *g*, 4 °C, 30 min), the supernatant was collected. 30 μl of supernatant was mixed with 90 μl 95% ethanol in a clear 96-well plate. The fluorescence intensity was measured at 620 nm excitation/680 nm emission on a microplate fluorescence reader and compared against a standard curve (serial dilutions of the stock dye solution in the concentration range of 0, 0.25, 0.5, 1, 5, 20 μg/mL). The Evans blue content was calculated and expressed as per gram of brain tissue.

### Stereotactic adeno-associated viruses (AAV) injection
10-week-old mice were used for AAV injection as described before with some modifications[37]. Briefly, mice were administered 0.1 mg/kg of buprenorphine (s.c.) subcutaneously and 0.2 mg/kg of dexamethasone (i.p.) before surgery. Subsequently, under continuous inhalational isoflurane anesthesia (5% for induction and 2% for maintenance with 66% O$_2$ and 33% N$_2$O), the mouse head was fitted and secured by blunt ear bars in a stereotactic apparatus (Robot stereotaxic, Neurostar) and the mouse eyes were covered by Bepanthen (Bayer, Leverkusen). After sterile cleaning and skin incision, viruses were injected at a rate of 0.1 μl min$^{-1}$ with a 10 μl Nanofil syringe (34 GA blunt needle, World Precision Instruments) into the striatum (1.1 mm anterior to bregma, ±1.2 mm from sagittal suture at a depth of 3 mm from the skull surface) or cortex (1.9 mm posterior to bregma, 1.5 mm from sagittal suture at a depth of 0.5 mm from the skull surface). The syringe was kept in place for 10 min after the injection was completed, to avoid liquid reflux. The skin was closed by simple interrupted sutures (non-absorbable, F.S.T.). Carprofen were administered once per day for three consecutive days and received tramadol hydrochloride in the drinking water (400 mg/L) after surgery. Mice were used for experiments 3 weeks post virus injection. Viruses used were: AAV2/5 GfaABC1D-GRAB$_{Ado}$ virus (2.3 × 10$^{12}$ genome copies/ml); AAV2/5 GFAP-hM4Di-mCherry virus (3.04 × 10$^{12}$ genome copies/ml); AAV2/5 GFAP-tdTomato virus (1.1 × 10$^{13}$ genome copies/ml), 0.8 μl for striatum, 0.5 μl for cortex, respectively.

### Craniotomy and window implantation
The skull of the mice was exposed and underwent a craniotomy followed by removal of the dura[68,69] in correspondence of the primary somatosensory cortex (SSp-ctx; Ø 3-4 mm, centered 1.5-2 mm posterior to bregma, 1.5 mm from the sagittal medial axis). The glass coverslip (Ø 3 mm, No. 1.5, Glaswarenfabrik Karl Hecht GmbH) and a custom-made holder for head restraining were fixed with dental cement (RelyXTM Unicem 2 Clicker, 3 M Deutschland GmbH). Animals received canonical post-surgical care; three-day habituation training and imaging was performed one week after cranial window implantation.

### In vivo imaging of extracellular adenosine level
In vivo imaging from GRAB$_{ado}$ virus injected mice was performed using a custom-made two-photon laser-scanning microscope (2P-LSM) setup. Recordings started one week after cranial window implantation to enable animal recovery and habituation according to adapted protocols without water restriction[68,69]. During image acquisition at different time points after an LPS challenge, awake mice were maintained head-fixed and placed into a smooth plastic tube to reduce vibration caused by the movement of the mice. In the drug administration experiments, the head-fixed mice received 1.5% isoflurane anesthesia delivered by means of a custom-made nose mask. Solutions of adenosine (0, 5, 10 or 20 mg/kg) and SR101 (20 mg/kg) were prepared in isotonic saline solution and administered after 200 frames of baseline recording (~88 s). Squared FOVs (512 ×512 μm) were chosen at a 60–80 μm depth from the surface and displaying a uniform virus expression with 910-nm laser. The laser power under the objective was

adjusted from 5 to 30 mW, depending on the focal plane depth. Single focal plane images were acquired with a 2.28 Hz frame rate (1.4 μs pixel dwell time) and a resolution of 512 × 512 pixel.

## Adenosine, adenosine analogue NECA, A1AR agonist CPA/CCPA, A1AR antagonist DPCPX treatment

To pharmacologically activate adenosine signaling, adenosine (5 mg/kg per body weight) (A9251, Sigma) was injected intraperitoneally every hour till 6 hours post first injection; NECA (1 mg/kg) (1691, Tocris) was injected intraperitoneally once at 0 hpi; CPA (1 mg/kg) (ab120398, abcam) was injected intraperitoneally once at 0 hpi; CCPA (0.3 mg/kg) (1705, Tocris) was injected intraperitoneally once at 0 hpi; DPCPX (1 mg/kg) (ab120396, Abcam) was injected intraperitoneally 4 hours after LPS (5 mg/kg) or Veh (1% DMSO in PBS) injection. To further amplify inflammatory response, CPA (1 mg/kg) was injected intraperitoneally 4 hours after LPS (1 mg/kg). Mice were analysed at 6 hours post injection.

## Primary astrocyte culture and drug treatment

Astrocytes were cultured from ctl mice at postnatal day 0 (P0) to P1 as previously described[70]. Briefly, hippocampi were dissected and treated with trypsin for 10 min at 37 °C. Tissues were dispersed through a cell strainer in culture medium (DMEM (Fisher Scientific) + 10% FCS (Fisher Scientific) + 0.1% Pen/Strep (Fisher Scientific) + 0.05% Mito Serum Extender (VWR) and centrifuged at 2300 rpm for 4 mins. Cells were plated into tissue culture flasks (ZHCL-Sarstedt) coated with collagen (1 mg/ml; Advanced BioMatrix) and kept in culture medium in a humidified 8.5% $CO_2$/91.5% air atmosphere at 37 °C. Confluent cultures (7–10 days) were shaken vigorously, the remaining astrocytes were trypsinized and plated in 6-well plates (ZHCL-Sarstedt). Purified astrocytes were used 5–7 d after plating until confluent. CCPA (30 nM) or CV1808 (50 nM) were added into the culture and incubated for 6 h. Subsequently, the cells were slightly rinsed with 1× PBS and homogenized in RLT buffer (Qiagen) for RNA extraction by RNeasy mini kit (74106, Qiagen).

## Reverse transcription-polymerase chain reaction (RT-PCR)

Deeply anesthetized mice were perfused with ice-cold PBS. Cortex and striatum were dissected in ice cold PBS using an adult brain reference atlas. Total RNA of cortex and striatum were isolated with the RNeasy mini kit (74106, Qiagen) according to the instructions and underwent 15-min on-column DNase digestion using an RNase-Free DNase Set (79254, Qiagen). cDNA was prepared with reverse transcriptase using an Omniscript RT kit (205113, Qiagen) following the standard protocol provided. RT-PCR was performed using EvaGreen (27490, Axon) kit with CFX96 Real Time System (BioRad). The standard two step-program was used: 95 °C for 10 min, then 40 cycles at 95 °C for 15 sec, 60 °C for 1 min. The expressions of *Gfap, Itgam, Pdgfra, Pdgfrb, Cxcl1, Cxcl10, Ccl2, Ccl5, Lcn2, Il6, Il1a, Il1b, Tnf* were measured. β-actin was performed as a control in all qPCR analysis. Primers were designed to work at approximately + 60 °C and the specificity was assessed by melt curve analysis of each reaction indicating a single peak. Primer used in this study are listed in Supplementary Table 1. Relative expression of all targeted genes was determined using the ΔΔCt method with normalization to *Actb* expression.

## Immunohistochemistry

Anesthetized mice were perfused with 1× PBS and 4% paraformaldehyde (PFA). Subsequently, the brain was dissected and post fixed in 4% PFA overnight (4 °C). Fixed brains were sliced into 35 μm thickness coronal sections in correspondence to the primary somatosensory cortex (AP −1.6 ± 0.5) and striatum (AP 0.5 ± 0.5), in PBS, using a Leica VT1000S vibratome and collected in 48-well culture plates containing 1× PBS. Vibratome sections were incubated for 1 h in blocking buffer (5% HS, 0.3% Triton X in 1× PBS) at room temperature. Sections were

incubated with primary antibodies, diluted in the blocking solution overnight at 4 °C. The next day, sections were washed 3 times (10 min, 1× PBS) and incubated for 2 h with secondary antibodies and DAPI diluted in blocking buffer at room temperature in the dark. Subsequently, the sections were washed and mounted in Shandon Immu-Mount (Thermo Scientific). For p-stat3 staining, slices in correspondence to the somatosensory cortex and striatum (AP −1.6 ± 0.5) was incubated in citrate buffer (pH = 6) at 95 °C for 5 mins as antigen retrieval. Primary and secondary antibodies are listed in Supplementary Table 1: key resources.

## Image acquisition and analysis

Whole brain slices were scanned with the automatic slide scanner AxioScan.Z1 (Zeiss, Oberkochen) as described before[71]. Cell counting from whole brain slices was manually performed using ZEN software (Zeiss, Jena). c-Fos intensity within Sox9+ and NeuN+ areas were automatically measured by using the machine learning function of the ZEN software. For the 3D reconstruction of microglia, confocal images were taken by LSM 710 and LSM880 confocal microscope (Zeiss, Oberkochen) with 1 μm intervals. Reconstruction of the microglial surface was performed using IMARIS (Version 9.6, Oxford Instruments) at the following settings: surface detail 0.700 μm (smooth); thresholding background subtraction (local contrast), diameter of largest sphere: 2.00. Next, the surface reconstruction was used as template for filament reconstruction with the following settings: detect new starting points: largest diameter 7.00 μm, seed points 0.300 μm; remove seed points around starting points: diameter of sphere regions: 15 μm. After filament reconstruction, individual data sets of Sholl analysis were exported into separate Excel files for further analysis. Image processing, three-dimensional reconstruction, and data analysis were performed in a blind manner regarding the experimental conditions.

## Ribosome immunoprecipitation (IP)

After perfusion with ice-cold HBSS, cortical regions samples were dissected from mouse brain and stored at −80 °C until use. Tissues were homogenized in ice-cold lysis buffer (50 mM Tris, pH 7.4, 100 mM KCl, 12 mM $MgCl_2$, 1% NP-40, 1 mM DTT, 1× protease inhibitor, 200 units/ml RNasin (N2518, Promega) and 0.1 mg/ml cycloheximide (C7698, Sigma) in RNase-free deionized $H_2O$) 10% w/v with homogenizer (Precellys 24, PeQlab). Then homogenates were centrifuged at 10,000 × g at 4 °C for 10 min to remove cell debris. Supernatants were collected and removed: 50 μl were used as input analysis. Anti-HA Ab (1:100, MMS-101R, Covance) was added to the supernatant with slow rotation at 4 °C. Meanwhile, Dynabeads Protein G (10004D, Thermo Fisher Scientific) were equilibrated to the lysis buffer by washing three times. After 4 h of incubation with HA Ab, 100 μl pre-equilibrated beads were added to each sample and incubated overnight at 4 °C. After 10-12 h, Dynabeads were washed with high-salt buffer (50 mM Tris, 300 mM KCl, 12 mM $MgCl_2$, 1% NP-40, 1 mM DTT, 1× protease inhibitor, 100 units/ml RNasin and 0.1 mg/ml cycloheximide in RNase-free deionized $H_2O$) three times for 5 min at 4 °C. At the end of the washing, beads were magnetized and 350 μl RLT buffer from RNeasy Micro Kit (74004, Qiagen) was added to the beads. RNA was extracted followed with manufacturer's instructions.

## Next-generation RNA sequencing

The library was prepared and sequenced by Novogene using the following methods. Preliminary quality control was performed on 1% agarose gel electrophoresis to test RNA degradation and potential contamination. Sample purity and preliminary quantitation were measured using Bioanalyser 2100 (Agilent Technologies, USA), which was also used to check the RNA integrity and final quantitation. For library preparation, the mRNA present in the total RNA sample was isolated with magnetic beads of oligos d(T)25. This method is known as polyA-tailed mRNA enrichment. Subsequently, mRNA was randomly

fragmented and cDNA synthesis proceeds using random hexamers and the reverse transcriptase enzyme. Once the synthesis of the first chain is finished, the second chain is synthesized with the addition of an Illumina buffer. Together with the presence of dNTPs, RNase H and polymerase I from E.coli, the second chain will be obtained by nick translation. The resulting products go through purification, end-repair, A-tailing and adapter ligation. Fragments of the appropriate size are enriched by PCR, where indexed P5 and P7 primers are introduced, and final products are purified. The library was checked with Qubit 2.0 and real-time PCR for quantification and bioanalyzer Agilent 2100 for size distribution detection. Quantified libraries were pooled and sequenced on Illumina Novaseq 6000 platform, according to effective library concentration and data amount.

The qualified libraries were sequenced next generation sequencing (NGS) based on Illumina's sequencing technology by synthesis (SBS)- detection by fluorescence of the nucleotide added during the synthesis of the complementary chain—and in a parallelized and massive way. The Novaseq6000 sequencing system was used to sequence the libraries. The strategy is paired end 150 bp (PE150).

## RNA-seq data processing
The RNA sequencing reads quality was assessed using FastQC (https://www.bioinformatics.babraham.ac.uk/projects/fastqc/). Reads were aligned to the GRCm38 Mus musculus genome using HISAT2 v2.0.5[72] with default parameters. Gene count matrix of each sample was generated by featureCounts v1.5.0-p3[73]. Downstream analysis was performed with the DEseq2 v1.20.0[74] package in R. Genes with normalized count below 10 were removed from downstream analysis. Significantly deregulated genes were identified by a false discovery rate lower than 0.05. Differential expression analysis between conditions was performed with the 'DESeq' function with default parameters. Log2 fold-change shrinkage was performed on the differential expression analysis result. Heatmaps of DEGs with $p^{adj}$ value < 0.05 in any time point were visualized with pheatmap v1.0.12 using Z-scores. A K-means algorithm was used for unsupervised clustering of gene sets with similarities involved in different time points post LPS injection as used in previous studies[45,75]. Selected gene set enrichment analysis of gene ontology (GO) and Kyoto Encyclopedia of Genes and Genomes (KEGG) pathway was performed with ClusterProfiler v3.8.1[76]. Transcription factor prediction was performed using Metascape v3.5.20230501[77,78] with the differentially expressed genes identified by DEseq2.

## Cytokine expression analysis
After perfusion with ice-cold PBS, the cortex and striatum were dissected from coronal brain slices (1 mm) and stored at −80 °C until further processing. For the cytokine array on brain homogenates, we used the Proteome Profiler Mouse Cytokine Array Kit (ARY006, R&D Systems) and Proteome Profiler Mouse XL Cytokine Array (ARY028, R&D Systems) following the manufacturer's instructions. Briefly, brain tissue was homogenized in PBS containing 1× protease inhibitor cocktail (05892970001, Roche). Protein concentration was measured using the bicinchoninic acid (BCA) assay kit (23225, Thermo Fisher Scientific). 200 μg protein lysis were used for each membrane. Membranes were blocked, incubated, and washed according to standard protocol. Four membranes were exposed to X-ray film (47410, Fuji) for 15 minutes at the same time. The intensity (pixel density) of each spot on membrane was quantified using Image J software (National Institutes of Health, Bethesda), corrected for background intensity, and normalized to control.

## Flow cytometry analysis of immune cells
Flow cytometry analysis of infiltrated immune cells into the brain was performed and defined as previously described with some modifications[79]. Adult mice brains were dissected after PBS/LPS injection and dissociated with the Adult Brain Dissociation Kit (Miltenyi Biotec, 130-107-677). The cell suspension was collected and washed with FACS buffer (1× PBS containing 3% fetal bovin serum). The prepared single cells were incubated with CD16/CD32 antibody (Miltenyi Biotec, 130-107-066) at 4 °C for 10 min to block Fc receptors. Afterwards, cells were stained with fluorophore-conjugated antibodies: CD3-APC-Cyanine7 (100221, BioLegend), CD11b-PerCP (101229, BioLegend), CD45-BV421 (103133, BioLegend), Ly6C-PE-Cyanine7 (128017, BioLegend) and Ly6G-APC (127613, BioLegend) at 4 °C for 30 min in the dark. After washing, cells were resuspended in FACS buffer for analysis by BD FACSVerse™ flow cytometer. The data were analyzed by Flowjo.

## Acute brain slice preparation
Acute brain slice preparation was performed as previously described[80]. Briefly, 11–13-week-old mice were anesthetized with isofluran (Abbvie, Ludwigshafen, Germany) and euthanized by cervical dislocation followed by decapitation. The brain was swiftly dissected and placed into ice-cooled, carbogen-saturated (5% $CO_2$, 95% $O_2$) cutting solution containing 87 mM NaCl, 3 mM KCl, 25 mM $NaHCO_3$, 1.25 mM $NaH_2PO_4$, 3 mM $MgCl_2$, 0.5 mM $CaCl_2$, 75 mM sucrose, and 25 mM glucose. 300 μm-thick sections in correspondence to the primary somatosensory cortex (SSp-ctx, AP −1.6 ± 0.6) were cut with a vibratome VT1200S (Leica Biosystems, Wetzlar, DE) using a 0.12 mm/s cutting speed and a 1.9 mm cutting amplitude. The procedure between decapitation and incubation was performed as fast as possible and in no longer than 10 min for optimal quality of the slice preparation. Slices were incubated for 30 min on a custom-made nylon-basket submersed in artificial cerebral spinal fluid (ACSF) containing 126 mM NaCl, 3 mM KCl, 25 mM $NaHCO_3$, 15 mM glucose, 1.2 mM $NaH_2PO_4$, 1 mM $CaCl_2$, and 2 mM $MgCl_2$ at 32 °C. Subsequently, brain slices were taken out of the water bath and placed to RT with continuous oxygenation before use.

## $Ca^{2+}$ imaging in hippocampal slices
For acute brain slices of 11–13-week-old GLAC$^{ct2/wt}$xA1AR$^{wt/wt}$xGCaMP3$^{fl/fl}$ mice and GLAC$^{ct2/wt}$xA1AR$^{fl/fl}$xGCaMP3$^{fl/fl}$ mice were prepared as described above. Individual slices were transferred to a self-made imaging chamber under a custom-made 2P-LSM and fixed by stainless steel rings with 1 mm-spaced nylon fibres (Harp Slice Grid HSG-5A; ALA Scientific Instruments Inc., Farmingdale, NY, USA). A constant flow of oxygenated perfusion solution continuously perfused the imaging chamber at a flow rate of 2–5 ml/min by a peristaltic pump LKB P-1 (Pharmacia LKB, Uppsala, SE). Squared field of views (FOVs, 170 ×170 μm) from the cortical layers II-III of the SSp-ctx were chosen at a depth ranging from 30 to 100 μm from the slice surface and displaying a uniform astroglial distribution. Focal CPA (0.1 mg/ml) and SR101 (0.1 mg/ml) application was performed by means of borosilicate glass pipettes (BF150-86-10, Sutter Instrument, Novato, CA, US) mounted on 3-axis micromanipulator units (LN Mini 25, Luigs & Neumann GmbH, Ratingen, DE) controlled by a SM5 Remote Control station (Luigs & Neumann GmbH, Ratingen, DE). Pipettes were pulled using a micro-pipette Puller P-97 (Sutter Instrument, Novato, CA, US) to obtain a 5 μm-opening tip (0.5-1 MΩ). Cortical astrocytic $Ca^{2+}$ activity was recorded upon drug application. Analysis of $Ca^{2+}$ imaging data was performed using the custom-made detection and analysis software MATLAB (MSparkles v. 1.8.18)[32] as previously described[31]. Shortly, fluorescence fluctuations at basal $Ca^{2+}$ concentrations ($F_0$) were computed along the temporal axes of each individual pixel using a polynomial fitting in a least-squares sense. The range projection of $\Delta F/F0$ was then used to identify local fluorescence maxima, serving as seed points for simultaneous, correlation-based growing of regions of activity (ROAs).

## Electrophysiology of brain slices (LTP)
LTP recordings from dorsal hippocampus were performed as previously described[80,81]. Briefly, acute hippocampal brain slices were

prepared as mentioned above. Afterwards, slices were transferred to the recording chamber and continuously perfused with oxygenated ACSF containing (in mM)1 MgCl2 and 2.5 CaCl2 at a flow rate of 2–5 mL/min. Field excitatory postsynaptic potentials (fEPSP) were recorded by a micropipette of 1–3 MΩ resistance filled with ACSF in CA1 of hippocampus by stimulating Schaffer collaterals of CA3 using stimulus isolator and a biopolar electrode (WPI). Picrotoxin (50 μM) was perfused in the bath to inhibit ionotropic γ-aminobutyric acid type A receptors (GABAARs). Stimulus duration was 200 μs, current injection was 30–80 μA. To evoke LTP, triple θ-burst stimulation (TBS3) was used. TBS consisted of 10 bursts (4 pulses each burst, 100 Hz) delivered at an interburst interval of 200 ms, and repeated once at 10 s. The stimulation intensity was adjusted to evoke ~30–60% of the maximum response. Waveform analysis was performed by Igor pro 6.3.7.2. The statistical analysis was conducted in Graphpad Prism. All experiments were conducted at RT.

### Behavioral test

To evaluate the behavioural deficits following the LPS challenge, we evaluated open field test and sucrose preference test following the methods from previous studies with modifications[82,83]. For the open field test, mice were placed at a random corner of the open field square (50 cm length × 50 cm width × 38 cm height) and positioned facing the wall. The movement of each mouse was recorded for 10 minutes. Open field test was performed before and after LPS challenge. Duration time in the center area (s), moved distance (cm) and speed (cm/s) were analyzed by EthoVision XT 11.5 (Noldus Technology). For the sucrose preference test, mice were single caged and habituated to the presence of two drinking bottles for 2 days. After the acclimatization, mice were presented to two drinking bottles: one containing 1% sucrose and the other tap water for 2 days in their cage. The positions of two drinking bottles were exchanged daily. Water and sucrose solution intake was measured daily by weighing the bottles. Sucrose preference was calculated as a percentage of the volume of sucrose intake over the total volume of fluid intake and averaged over the 2 days of testing. Sucrose preference test was performed before and after LPS challenge.

### In vivo activation of hM4Di

CNO (half-life = ~0.5 h) was shown to activate hM4Di to enhance astrocytic Gi signaling for ~1.5–2 h in vivo[84]. Therefore, three weeks after appropriate microinjection of AAV2/5-GFAP-hM4Di-mCherry or AAV2/5-GFAP-tdTomato into the striatum or cortex, CNO was administered two times to animals by intraperitoneal injection (2 mg/kg; dissolved in saline) at 2 hours and 4 hours post LPS challenge, respectively. 6 hours or 24 hours after LPS challenge, animals were used for behavior tests or immunohistochemistry.

### Statistics and reproducibility

The statistical analyses of all data were performed with GraphPad Prism 9.5.1 statistical software (GraphPad, San Diego). For all immunostainings, two randomly selected brain slices of each mouse were studied. In addition, for the analysis of Iba1, more than 8 ROIs per mouse was analyzed. For normally distributed datasets, unpaired $t$ tests, paired $t$ tests (for studies of behavior), one-way ANOVA, and two-way ANOVA were used (indicated in each figure legend), while the Kruskal–Wallis test was used for non-normally distributed datasets. $P$ values are indicated in the figures and legends. For the in vivo experiments, each data point represents the data obtained from a single mouse (except for electrophysiology). The total mouse numbers are indicated in the figure legends and source data. For electrophysiology, each data point refers to a slice and the mouse number is indicated in the figure legends. Normal distributed datasets are shown as mean ± SEM. else are shown as the median ± IQR are indicated as thick and thin dashed lines, respectively. $p$ value of ≤0.05 was considered statistically significant.

### Reporting summary

Further information on research design is available in the Nature Portfolio Reporting Summary linked to this article.

## Data availability

The data supporting the findings of this study are available from the corresponding authors upon request. The RNA sequencing data have been deposited in the GEO database GSE248275. Source data are provided with this paper.

## Code availability

Custom-written R codes have been made available on Github [https://github.com/qilinguo1005/Adenosine-triggers-early-astrocyte-reactivity.git].

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

## Acknowledgements

We thank Daniel Schauenburg and colleagues for excellent animal husbandry. We thank Dr. Yulong Li (Peking University) for providing the AAV-GRAB$_{Ado1.0}$ plasmids; Dr. Amit Agarwal (University of Heidelberg) for providing Rosa26-GCaMP3 mice, Dr. André Zeug (Hannover Medical School) for providing the AAV2/5-GFAP-tdTomato virus. This work was supported by grants from the Deutsche Forschungsgemeinschaft (HU 2614/1-1 (Project No. 462650276) to W.H., Sino-German joint project KI 503/14-1 to W.H. and F.K., SPP 1757, SFB 894, SFB 1158 to F.K.); the Fritz Thyssen Foundation (10.21.1.021MN) and the Medical faculty of the University of Saarland (HOMFOR2016, HOMFORexzellent2017, HOMFOR2024 Anschubfinanzierung) to W.H.; the BMBF (EraNet-Neuron BrIE) and the European Commission (EC-H2020 FET ProAct Neurofibres) to F.K. The flow cytometry platform is funded by INST 256/423-1 FUGG from DFG. The collaborative work of W.H., F.K., and S.B. was supported by the Mobility Project (M-0679) from the Sino-German Research Center for Research Promotion. Q.L. was supported by a PhD student fellowship of the Chinese Scholarship Council.

## Author contributions

W.H. conceptualized the project and designed experiments with input from F.K. and Q.G. W.H. and F.K. supervised the study and ensured the coordination and resource support of the project. Q.G. performed RiboTag immunoprecipitation, immunohistochemistry, immunoblot, qPCR, confocal imaging and analysis, RNA-seq analysis, and behavioral tests. Q.G. and D.G. performed an AAV injection. Q.G., D.G., and W.H. carried out in vivo 2P-LSM imaging. D.G. performed ex vivo slice imaging and all 2P-LSM imaging analysis. N.Z. performed LTP recordings. Q.G., T.G., and M.R.M. performed plasma adenosine concentration measurements. Q.L. contributed to immunohistochemistry and behavioral tests. L-P.F., X.B., and S.B. contributed to sample collection and qPCR. H.Z. and R.Z. performed flow cytometry, and N-O.K. carried out the primary astrocyte cultures. A.S. performed AxioScan imaging. Q.G. analyzed data and generated figures. W.H. and Q.G. wrote the manuscript with input from all the other authors. F.K. revised the manuscript.

## Funding

## Competing interests

The authors declare no competing interests

## Additional information

¹Molecular Physiology, Center for Integrative Physiology and Molecular Medicine (CIPMM), University of Saarland, 66421 Homburg, Germany. ²Center for Gender-specific Biology and Medicine (CGBM), University of Saarland, 66421 Homburg, Germany. ³Institute of Anatomy and Cell Biology, University of Saarland, 66421 Homburg, Germany. ⁴Biophysics, CIPMM, University of Saarland, 66421 Homburg, Germany. ⁵Molecular Neurophysiology, CIPMM, University of Saarland, 66421 Homburg, Germany. ⁶Department of Experimental and Clinical Toxicology, Institute of Experimental and Clinical Pharmacology and Toxicology, Center for Molecular Signaling (PZMS), University of Saarland, 66421 Homburg, Germany. ⁷Institute for Regenerative Medicine, Shanghai East Hospital, Frontier Science Center for Stem Cell Research, School of Life Sciences and Technology, Tongji University, 200092 Shanghai, China. ✉e-mail: frank.kirchhoff@uks.eu; wenhui.huang@uks.eu

