## [Peer Review File · Nature Communications]

Adenosine triggers early astrocyte reactivity that provokes microglial responses and drives the pathogenesis of sepsis-associated encephalopathy in miceREVIEWER COMMENTS

Reviewer #1 (Remarks to the Author):

This MS reports novel findings about early astrocytic reactivity driving via adenosine A1Rs sepsis-associated encephalopathy. The idea is interesting, the approaches by various experimental methods is impressive, and the data are of good quality. However, I guess it should be pointed out with greater emphasis that traditionally A2A and A3Rs are believed to be involved in immunological changes in the CNS, and the involvement of the A1R is rather unexpected. Furthermore, I observed that in some experiments the n-numbers are rather low. 3-4 similar experiments are questionable for most statistical evaluation. Many tests do not work at a such low number of independent variables. Otherwise I have only a few remarks.

Remarks:

1. L.105. The high and low dose of LPS should be already defined here (5 and 1 mg/kg, respectively).
2. L.106. "hpi" means probably "hour post injection".
3. L. 108. It should be written expressis verbis that adenosine does not pass the BBB under normal conditions, and therefore its disruption is needed for this event.
4. L.143. It should be stated here that adenosine was determined in the blood, and in two areas of the brain (cortex and striatum). By the way which cortex was chosen for determinations.
5. L.160. It astonishes that while the half-life of adenosine is about 5 h, and the half-life of NECA is only about 0.5 h, NECA was injected only a single time in contrast to adenosine which was injected 6-times.
6. L.159. The doses of NECA, CPA and DPCPX should be mentioned already here not only in the Methods section.
7. P.8. It appears that in Fig.3 the panels E-H are not discussed in the Results section.
8. P.11, last para. The open field can be used to demonstrate depressive-like behavior but the reduced running path is a rather non-specific marker. It would be better to show the shortening of the time spent in the center areas of the open field apparatus. This is widely accepted to indicate anxiolysis/depression.
9. L.371. The half-life of CNO is about 0.5 h, but more important is how long the activation/inhibition of G-proteins holds on.
10. L.578. Why was dexamethasone applied before surgery? That would increase the chance of bacterial infection.
11. L.744. I guess that slices of the striatum were similar to those of the cerebral cortex.
12. Fig. 2. "bACT" stands supposedly for β -actin.
13. Fig. 4E. How was statistical significance calculated for these graphs?

Reviewer #2 (Remarks to the Author):

In their manuscript, Guo et al. demonstrate a novel role of adenosine as a trigger of early astrocyte reactivity in a mouse model of systemic LPS-induced inflammation. The injection of adenosine, adenosine analogues, or A1AR agonists was able to mimic the inflammatory response of astrocytes observed at early timepoints following LPS-induced neuroinflammation. The authors use a conditional A1AR ablation approach to demonstrate that astrocyte-specific sensing of adenosine through A1AR contributes to the pathological effects of reactive astrocytes, including the pro-inflammatory activation of microglia, BBB leakage, and neuronal dysfunctions. Finally, the authors postulate in contrast to the general belief that astrocyte reactivity in the LPS-induced neuroinflammation model is secondary to microglial activation, that astrocytes boost microglia activation during early stages of systemic inflammation.

These findings shed light on the role of adenosine in the context of sepsis-associated encephalopathy and are thus of high scientific, and potentially translational value, since they provide important insight into sepsis-associated encephalopathy. Thus, it is of great importance

and relevance to the field. However, as a major point of critique, the study lacks a translational perspective that aims to implement these observations into a clinical setting. Are Adenosine levels also elevated in the blood of sepsis patients? Does pharmaceutical blockage of this pathway ameliorate sepsis-induced encephalopathy?

Additional points to be addressed in a revised version of the manuscript are listed below:

Page 5:

1. Line 12: it is unclear why EB is injected intraperitoneally. In the methods section (and Figure 1A) it is described to be administered by injection into the tail vein. Please clarify.
2. Line 15: it is unclear how changes in adenosine plasma availability relate to BBB permeability. Use of the GRABAdo1.0 adenosine sensor without celltype specificity could provide a more convincing association between BBB permeability and adenosine availability in the CNS.
3. Line 19: Additional activation markers like CD69, TMEM119, P2YR12 for microglia should be used to determine their activation status. Furthermore, systemic inflammation induced by LPS and the increase in BBB permeability leads to the infiltration of immune cells, already at early timepoints (including other CD11b+ cell types). Using a flow cytometry based (sorting) approach would help to assess the activation of the cell types described here.
4. Line 22: It is unclear whether the inflammatory transcripts Figure S1B/C were assessed in bulk tissue or astrocytes.

Page 6:

5. Line 3: It would be informative to assess whether different doses of adenosine injection can impair BBB functions and EB extravasation (as performed in the initial experiment).
6. Line 13: The dose and timepoint used for the assessment remain unclear. Please clarify.
7. Lines 22-26: How can the differences in gene expression following Adenosine vs. NECA injection be explained? Are there different specificities?

Page 7:

8. While the initial observation was made in a LPS-induced sepsis model, the effect of adenosine on astrocyte reactivity may be a common feature of various forms of neuroinflammation. The use of the A1AR KO/RiboTag model in another neuroinflammatory mouse model (e.g. EAE) would strengthen the story significantly.
9. The authors demonstrate solely the importance of A1AR signaling for astrocyte reactivity. Using primary cultures of astrocytes (and potentially additional cell types like oligodendrocytes, microglia, and pericytes) in combination with individual adenosine receptors agonists/antagonists would strengthen the story.
10. Line 17: It is surprising that the inability to sense adenosine (through A1AR) in other cell types (in particular microglia) would not affect inflammatory processes in the CNS.
11. Lines 22-25: It is unclear why the authors used cFOS and Sox9 as proxy for astrocyte activation, rather than e.g. GFAP reactivity or NfκB signaling (as performed for the assessment of microglia activation).

Page 9:

12. Astrocytes modulate microglia responses through a variety of signaling pathways, of which NfκB signaling is one of many. For the claim that astrocytic-A1AR deficiency reduces microglia activity, additional parameters and functional properties of microglia should be assessed.

Page 11:

13. Lines 1-4: The effects of A1AR deficiency in astrocytes on BBB properties and neutrophil migration should be further elucidated by analyzing their production of chemoattractants and growth factors on a protein level.
14. Including DPCPX or other adenosine receptor antagonists in the behavioral test would provide additional translational relevance of the findings (despite being celltype unspecific).

Reviewer #3 (Remarks to the Author):

I co-reviewed this manuscript with one of the reviewers who provided the listed reports as part of the Nature Communications initiative to facilitate training in peer review and appropriate recognition for co-reviewers.

Reviewer #4 (Remarks to the Author):

The manuscript provides a detailed analysis of LPS's early effects on adenosine and astrocytes, with an emphasis on adenosine dynamics following LPS intraperitoneal injections and the molecular, cellular, and behavioral outcomes in cKO mice of A1R in astrocytes. The introduction of new molecular mechanisms in astrocyte responses to systemic inflammation and its implications for mouse phenotypes is both novel and insightful. While recognizing the importance of this research, I have several recommendations for improvement:

Major suggestions:

Interpretations

1. The variation in astrocyte calcium signaling before CPA application between control and cKO groups in Figure S2 needs further analysis and discussion.
2. The observed reduction in Ly6B+ neutrophils in cKO mice warrants further exploration to determine if it's due to less BBB impairment or reduced cytokines.
3. In light of recent consensus papers published at Nature Neuroscience in 2021 and at Neuron in 2022, terms like "astrocyte reactivity" and "microglial activation" should be avoided. Focus on specific molecular changes, such as "elevation of xx levels". Please note that Fos family proteins in astrocytes have been shown as downstream elements of the GPCR pathway signaling, but not established as activation markers.

Clarifications

4. Do GRAB adenosine sensors expressed on astrocyte membranes detect adenosine that is released from, or accumulates on, astrocytes? This inquiry stems from the observed incongruity between the peak of BBB permeability at 6 hours following an LPS i.p. injection and the progressive intensification of GRABAdo signals from 6 to 24 hours post-injection.
5. The manuscript should explain the rationale behind using K-means clustering, including its specific application and objectives.
6. The authors are encouraged to provide a more detailed explanation for their choice of brain regions under investigation, particularly the selection of the striatum in Figure 2C and the hippocampus in Figures 5D-F.
7. The manuscript mentions having data on A1AR cKO in various cell types but does not present it. Including this data at least to reviewers could be critical in establishing astrocyte specificity.

Data presentation

8. The authors should double check what the whiskers of each data point mean in Figure 1 B, C and F and define in figure legends.
9. The layout of data in Figure 2 is challenging to interpret. Using heat plots might enhance clarity for readers.
10. The heatmaps in Figures 3 and S3 lack clarity, particularly regarding what each color represents.

11. In Figure 3D, the y-axis labeling in the heatmap is inadequate, making it extremely difficult to identify specific genes. This lack of clarity hinders the interpretation of gene alterations in cKO mice.

12. The authors may consider relocating Figure 4G-H panels to follow Figure 3 for a more coherent presentation, as they seem more relevant to it. Also, integrating the analysis of microglial phenotypes with Figure 5 (cellular and behavioral consequences of cKO) could be beneficial.

Minor points:

1. The authors should ensure to describe "Adora1 cKO", not just "cKO", is explicitly defined in all figures and legends for clarity.

2. The authors need to furnish comprehensive details regarding the reagents and equipment used, including catalog numbers and lot numbers where applicable, to enable reproducibility of the results.

3. The references to Nagai et al., 2019a and 2019b seem to be the same. It might be more accurate to cite Nagai et al., 2021 from Neuron.

point-by-point response to the reviewers' comments

Reviewer #1 (Remarks to the Author):

This MS reports novel findings about early astrocytic reactivity driving via adenosine A1Rs sepsis-associated encephalopathy. The idea is interesting, the approaches by various experimental methods is impressive, and the data are of good quality. However, I guess it should be pointed out with greater emphasis that traditionally A2A and A3Rs are believed to be involved in immunological changes in the CNS, and the involvement of the A1R is rather unexpected. Furthermore, I observed that in some experiments the n-numbers are rather low. 3-4 similar experiments are questionable for most statistical evaluation. Many tests do not work at a such low number of independent variables. Otherwise I have only a few remarks.

Author response: *Thank you for your very positive comments on our work and for the suggestion to emphasize that unlike A2 and A3 ARs, traditionally A1AR function in regulating neuroinflammation is largely unknown. We have added this point to the Introduction of the revised manuscript to highlight the novelty of our work. In addition, we think “3-4 similar experiments” were referred to the immunohistochemical analysis of reactive astrocytes and microglia as well as activated neurons after LPS challenge, therefore we performed more experiments. Increasing n, however, did not change the statistical significance, the biological impact remained the same (Fig. 3; Fig. 6a-f; Fig. 7a-d).*

Remarks:

1. L.105. The high and low dose of LPS should be already defined here (5 and 1 mg/kg, respectively).

Author response: *We defined the high and low dose of LPS here as suggested.*

2. L.106. “hpi” means probably “hour post injection”.

Author response: *We have corrected it as suggested.*

3. L. 108. It should be written expressis verbis that adenosine does not pass the BBB under normal conditions, and therefore its disruption is needed for this event.

Author response: *We have clarified this point in the conclusion of this section as suggested: “Taken together, we determined the dynamic changes of adenosine levels in the blood and somatosensory cortex post peripheral LPS challenge. Although the potential sources of the increased extracellular adenosine in the brain after LPS treatment remain unidentified and there is a lack of evidence from previous studies showing adenosine could pass the BBB under normal conditions, our current results strongly suggest that peripheral LPS injection induced a rapid*

increase of plasma adenosine which may contribute to the elevated extracellular adenosine level in the brain as well as a rapid neuroinflammatory response in the early phase of the systemic inflammation with a disruption of BBB function.”

4. L.143. It should be stated here that adenosine was determined in the blood, and in two areas of the brain (cortex and striatum). By the way which cortex was chosen for determinations.

Author response: *Thank you for this suggestion. We injected AAV-GRAB_{Ado1.0} to the somatosensory cortex for live imaging of extracellular adenosine levels. However, due to the depth of the striatum, it is difficult to perform the live imaging for this brain region. Nevertheless, previous studies using intracerebral recordings of adenosine biosensors in the rat hypothalamus suggest that systemic inflammation could increase extracellular adenosine levels in the ventral brain¹. In the revised manuscript, we stated these points as you suggested and cited previous studies accordingly to support our findings.*

5. L.160. It astonishes that while the half-life of adenosine is about 5 h, and the half-life of NECA is only about 0.5 h, NECA was injected only a single time in contrast to adenosine which was injected 6-times.

Author response: *We guess you may have overlooked these sentences. Actually we wrote “We injected adenosine several times due to its short lifetime in the blood (~ 1 h)”, and “we injected NECA (1 mg/kg, a non-selective adenosine analogue, half-life = ~ 5 h)”.*

6. L.159. The doses of NECA, CPA and DPCPX should be mentioned already here not only in the Methods section.

Author response: *We agree and added the dose information accordly as suggested.*

7. P.8. It appears that in Fig.3 the panels E-H are not discussed in the Results section.

Author response: *Thank you for pointing out this part. In the revised manuscript, we discussed these panels together with the results of Ca²⁺ imaging as suggested by Reviewer 4 for potential intracellular mechanisms of A1AR signaling enhancing the inflammatory response of reactive astrocytes.*

8. P.11, last para. The open field can be used to demonstrate depressive-like behavior but the reduced running path is a rather non-specific marker. It would be better to show the shortening of the time spent in the center areas of the open field apparatus. This is widely accepted to indicate anxiolysis/depression.

Author response: *Thank you for your suggestion. LPS-induced sepsis generates severe sickness of the mice which peaks at about 6 hpi and leads to post-sepsis complications such as*

fatigue, muscle weakness, behavioural deficits (e.g., depression/anxiety-like behavior). Therefore, we used the total distance travelled in the open-field test (OFT) as a general readout for the post-sepsis status. In the revised manuscript, we reanalyzed the data of the OFT. Please note a few data points in the previous version were from mice initially positioned in the middle of the open-field arena for the test which were done at the beginning of this project. Obviously, these results are not proper for analysis of the time spent in the central area because these mice barely move after LPS injection. Therefore, we replaced them with new results from mice initially positioned in the corner as we did to all other mice used in the following OFT. We showed that at 24 hpi the Adora1 cKO mice walked significantly more than the ctl mice. However, LPS heavily decreased the time spent in the central area of both groups, though the Adora1 cKO mice showed a tendency of increasing the time (Fig. 7h-j). Concerning other aspects of the sickness such as fatigue at 24 hpi may influence the interpretation of the time spent in the central area, we used the sucrose-preference test to evaluate the depression-like behaviour during 24-72 hpi more specifically (Fig. 7k). To describe these results more precisely, we rephased the text in the revised manuscript accordingly.

9. L.371. The half-life of CNO is about 0.5 h, but more important is how long the activation/inhibition of G-proteins holds on.

Author response: In recent studies, CNO was shown to activate hM4Di to enhance astrocytic Gi signaling for ~1.5 - 2 h in vivo^{2,3}. We have added this information to the Materials and Methods.

10. L.578. Why was dexamethasone applied before surgery? That would increase the chance of bacterial infection.

Author response: According to the animal welfare protection rules of the local government of Saarland in Germany, dexamethasone treatment is required for animals after surgeries to reduce the stress response during the operation and reduces cerebral edema. We obeyed to the rules.

11. L.744. I guess that slices of the striatum were similar to those of the cerebral cortex.

Author response: Here it shows the brain slices from the hippocampal level prepared for LTP recordings and Ca²⁺ imaging. We added the coordinate information of collecting brain slices for different brain regions in the Immunohistochemistry section.

12. Fig. 2. “bACT” stands supposedly for β-actin.

Author response: Yes, bACT was used for β-actin. Now we have corrected “bACT” to the gene name “Actb”.

13. Fig. 4E. How was statistical significance calculated for these graphs?

Author response: *We used two-way ANOVA as used in previous studies. This information has been stated in the figure legend.*

Reviewer #2 (Remarks to the Author):

In their manuscript, Guo et al. demonstrate a novel role of adenosine as a trigger of early astrocyte reactivity in a mouse model of systemic LPS-induced inflammation. The injection of adenosine, adenosine analogues, or A1AR agonists was able to mimic the inflammatory response of astrocytes observed at early timepoints following LPS-induced neuroinflammation. The authors use a conditional A1AR ablation approach to demonstrate that astrocyte-specific sensing of adenosine through A1AR contributes to the pathological effects of reactive astrocytes, including the pro-inflammatory activation of microglia, BBB leakage, and neuronal dysfunctions. Finally, the authors postulate in contrast to the general belief that astrocyte reactivity in the LPS-induced neuroinflammation model is secondary to microglial activation, that astrocytes boost microglia activation during early stages of systemic inflammation.

These findings shed light on the role of adenosine in the context of sepsis-associated encephalopathy and are thus of high scientific, and potentially translational value, since they provide important insight into sepsis-associated encephalopathy. Thus, it is of great importance and relevance to the field. However, as a major point of critique, the study lacks a translational perspective that aims to implement these observations into a clinical setting. Are Adenosine levels also elevated in the blood of sepsis patients? Does pharmaceutical blockage of this pathway ameliorate sepsis-induced encephalopathy?

Author response: *We are very grateful for your positive evaluation of our work. Regarding the concern on the translational perspective of our work, we would like to point out that previous studies detected increased plasma adenosine in septic patients⁴. Furthermore, volunteers receiving low dose of LPS also showed an elevated plasma adenosine level⁵. Indeed, these prior results encouraged us to investigate the potential role of adenosine mediating systemic inflammation signal to the CNS and we have cited these works in the manuscript accordingly. In addition, we already showed evidence that administration of the A1AR antagonist DPCPX at the early phase of systemic inflammation induced by LPS injection could reduce the inflammatory response of microglia (less nuclear p65⁺ microglia), inhibit the expression of inflammation-related cytokines/chemokines, and inhibit the aberrant neuronal activity (less c-Fos expression in neurons) (Fig. 2C; Supplementary Fig. 2d, e, I) . In the revised manuscript, we provide additional results demonstrating the DPCPX treatment could ameliorate the behavioural deficits of mice one day post the LPS injection (Supplementary Fig. 10). Therefore, we believe that our work highlights the*

translational potential of A1AR signaling as a therapeutic target for sepsis-induced encephalopathy.

Additional points to be addressed in a revised version of the manuscript are listed below:

Page 5:

1. Line 12: it is unclear why EB is injected intraperitoneally. In the methods section (and Figure 1A) it is described to be administered by injection into the tail vein. Please clarify.

Author response: *Thank you for pointing out this mistake. EB was injected i.v. for Fig. 1d and supplementary Fig. 8d). In the revised manuscript, we provided a new result (Fig. 1c) showing i.p. injection of adenosine could increase the EB extravastion in the brain. For this new experiment, EB was injected i.p. 10 min prior to the adenosine treatment. We have provided the corrected information in the Methods.*

2. Line 15: it is unclear how changes in adenosine plasma availability relate to BBB permeability. Use of the GRABAdo1.0 adenosine sensor without celltype specificity could provide a more convincing association between BBB permeability and adenosine availability in the CNS.

Author response: *Prior studies have demonstrated that activation of endothelial A1 or A2a adenosine receptors by peripheral injection of adenosine analogues (e.g. NECA, CCPA) could directly open the BBB⁶. In the revised manuscript, we also showed that i.p. injection of adenosine could increase the BBB permeability (Fig. 1c). It is well known that the astrocytic end-foot is an essential component of the BBB as interface between the brain parenchyma and the blood vessel. Recent in vivo studies further showed every astrocyte is wrapping up at least one blood vessel by extending the end-foot⁷. Therefore, expressing the GRABado1.0 sensor in astrocytes is ideal for our current study focusing on peripheral and central adenosine levels. On the other hand, it is difficult to express GRABado1.0 without celltype specificity because AAVs or other types of virus tools that can efficiently infect all CNS cells including oligodendrocyte lineage cells and microglia are not yet available.*

3. Line 19: Additional activation markers like CD69, TMEM119, P2YR12 for microglia should be used to determine their activation status. Furthermore, systemic inflammation induced by LPS and the increase in BBB permeability leads to the infiltration of immune cells, already at early timepoints (including other CD11b+ cell types). Using a flow cytometry based (sorting) approach would help to assess the activation of the cell types described here.

Author response: *Thank you for this very constructive suggestion. Of note, this remark is similar to remark No.12, thereby we combined our responses to both remarks here. In the revised manuscript, we provide results of a P2ry12 expression analysis by qPCR in the brain after LPS*

injection. We found that at 6 hpi, P2ry12 expression levels were downregulated in both *ctl* and *Adora1 cKO* mice. However, at 24 hpi the expression of P2ry12 in *Adora1 cKO* mice recovered to the healthy level while in *ctl* mice P2ry12 expression still significantly decreased (Fig. 6j). In addition, we performed CD68 (a marker for microglial lysosome) immunostaining to assess the phagocytosis of reactive microglia. We showed that at both 6 and 24 hpi, the size of CD68⁺ lysosome volume was significantly reduced in the *Adora1 cKO* mice (Fig. 6e). Taken together, our results suggest astrocytic-A1AR deficiency results in a less reactive state of microglia post LPS injection. According to this reviewer's suggestion, we performed flow cytometry to analyze the infiltration of peripheral immune cells at the early phase of the systemic inflammation post LPS injection. We found less infiltration of peripheral T cells, neutrophils and monocytes in *Adora1 cKO* mice at 6 hpi (Supplementary Fig. 8e-g).

4. Line 22: It is unclear whether the inflammatory transcripts Figure S1B/C were assessed in bulk tissue or astrocytes.

Author response: *These results were generated from the bulk tissue of the cortex. We stated this information in the text and also added this information to the figure legend of Supplementary Fig. 1.*

Page 6:

5. Line 3: It would be informative to assess whether different doses of adenosine injection can impair BBB functions and EB extravasation (as performed in the initial experiment).

Author response: *Thank you for this suggestion. In the revised manuscript, we added the results of the EB extravasation upon peripheral adenosine administration (Fig. 1c). Our results are in line with previous studies showing adenosine analogues could directly open the BBB⁶.*

6. Line 13: The dose and timepoint used for the assessment remain unclear. Please clarify.

Author response: *We clarified the dose and timepoint in the Results and Methods sections of the revised manuscript.*

7. Lines 22-26: How can the differences in gene expression following Adenosine vs. NECA injection be explained? Are there different specificities?

Author response: *Although NECA is a non-selective agonist of adenosine receptors, its intrinsic pharmacological properties are differed from adenosine. For example, it has been demonstrated that adenosine shows the highest affinity to A1 and A2a ARs, but significantly lower affinity to A2b and A3 ARs. However, NECA shows the high affinity to A1, A2a, and A3 ARs, but relatively lower affinity to A2b AR^{8,9}. In addition, previous studies suggest that A2 and A3 ARs signaling, particularly in microglia, contribute to neuroinflammation. Therefore, it is conceivable that the*

global neuroinflammatory responses to respective adenosine and NECA injections are not fully identical. Nevertheless, our results strongly suggest that peripherally administered adenosine can evoke neuroinflammation in the brain. Furthermore, specific deletion of A1ARs in astrocytes *in vivo* could significantly reduce such neuroinflammatory response by inhibiting the upregulation of numerous inflammation-related factors.

Page 7:

8. While the initial observation was made in a LPS-induced sepsis model, the effect of adenosine on astrocyte reactivity may be a common feature of various forms of neuroinflammation. The use of the A1AR KO/RiboTag model in another neuroinflammatory mouse model (e.g. EAE) would strengthen the story significantly.

Author response: *We agree that it is very necessary and important to investigate the astrocyte reactivity via A1AR signaling in other disease models such as EAE, which we also suggested in the Discussion. However, it is well known that astrocytes show highly context-dependent reactivity in different diseases¹⁰. Therefore, we would like to focus on the role of astrocytic A1ARs in SAE for the current manuscript. Another fact blocking us to switch our study to another disease model for now is that we will have to apply for a new animal experiment license from the local government which will last at least 6-8 months. Therefore, more experiments using different disease models are on our list but have to be performed in the future.*

9. The authors demonstrate solely the importance of A1AR signaling for astrocyte reactivity. Using primary cultures of astrocytes (and potentially additional cell types like oligodendrocytes, microglia, and pericytes) in combination with individual adenosine receptors agonists/antagonists would strengthen the story.

Author response: *As suggested, in the revised manuscript we provide new results from primary astrocytes treated by the CCPA (A1AR agonist) or CV1808 (non-selective A2AR agonist). Although it is widely accepted that *in vitro* experiments cannot fully mimic the complexity of *in vivo* models, we could still detect the upregulation of Cxcl1, Cxcl10, and Ccl5 in primary astrocytes triggered by CCPA. However, no significant alterations of the tested inflammation-related genes were detected in primary astrocytes triggered by CV1808 (Supplementary Fig. 5).*

10. Line 17: It is surprising that the inability to sense adenosine (through A1AR) in other cell types (in particular microglia) would not affect inflammatory processes in the CNS.

Author response: *In the revised manuscript, we added the results from microglia-A1AR deficient mice and OPC/pericyte-A1AR deficient mice upon challenges by CCPA (A1AR agonist). We found unaltered expression levels of inflammation-related factors in microglia-A1AR deficient mice,*

which is line with recent transcriptomic profiling studies revealing that A1AR expression in microglia is much lower than in astrocytes and oligodendrocyte lineage cells^{11,12}. In addition, in OPC/pericyte-A1AR mice, we found only Ccl2 expression was inhibited while Ccl5 expression is slightly upregulated, suggesting OPCs/pericytes may contribute to neuroinflammation in response to CCPA (Fig. 2e, Supplementary Fig. 2j, k). However, compared to the effect of inhibiting a broader spectrum of inflammation-related factors in astrocyte-A1AR deficient mice (Fig. 2e, Supplementary Fig. 2i), we still can conclude that A1ARs in astrocytes rather than in other glial cell types play a major role in inflammatory response to peripheral CCPA challenge.

11. Lines 22-25: It is unclear why the authors used cFOS and Sox9 as proxy for astrocyte activation, rather than e.g. GFAP reactivity or NfκB signaling (as performed for the assessment of microglia activation).

Author response: As an immediate-early gene, c-Fos has been shown to be transiently upregulated in astrocytes at the early phase of the peripheral LPS injection model in previous studies and also in the our current study¹³. In EAE model, c-Fos has been used to identify a subset of so-called immediate-early astrocytes which promote neuroinflammation^{14,15}. Furthermore, as mentioned by Reviewer 4 GPCR signaling (including Gq and Gi) has been shown to induce c-Fos expression in astrocytes^{16,17}. Since A1AR is a Gi protein-coupled receptor, we thereby used c-Fos to indicate the reactivity of astrocytes toward elevated extracellular adenosine levels in the LPS model. GFAP is not an ideal marker for early reactive astrocytes because there is a discrepancy between the mRNA and protein level of GFAP (i.e., the upregulation of GFAP protein starts in the late phase after 24 hpi)¹⁰. We found that it seems difficult to gain clear immunoreactive signals of different co-factors of NFκB (e.g., p65, p50) according to our own results and others' reports¹⁸. This may be attributed to its relatively low expression level in astrocytes compared to in microglia. To better evaluate the inflammatory response of astrocytes post LPS injection in the revised manuscript, we further studied the activation of STAT3 (a pivotal inducer of inflammatory response) in astrocytes by analysing the p-STAT3 immunoreactivity (Fig. 3e-h). We used Sox9 as an astrocytic marker to identify astrocytes which facilitates the analysis of astrocyte reactivity in terms of expression of c-FOS or p-STAT3.

Page 9:

12. Astrocytes modulate microglia responses through a variety of signaling pathways, of which NfκB signaling is one of many. For the claim that astrocytic-A1AR deficiency reduces microglia activity, additional parameters and functional properties of microglia should be assessed.

Author response: Please see our response to remark No. 3.

Page 11:

13. Lines 1-4: The effects of A1AR deficiency in astrocytes on BBB properties and neutrophil migration should be further elucidated by analyzing their production of chemoattractants and growth factors on a protein level.

Author response: *We used a cytokine array to test the expression of different inflammation-related genes at the protein level in Adora1 cKO and ctl mice post LPS injection. We could show that the expression of several chemoattractants (e.g., CCL2, CCL5, CXCL1, CXCL10, ICAM-1) of peripheral immune cells as well as many other inflammation-related factor (e.g., IL-1 α , Lcn2) were inhibited in the Adora1 cKO mice (Fig. 5; Supplementary Fig. 9). We agree it would be ideal to detect them in astrocytes in vivo by immunohistochemistry. Therefore, we made many efforts to test several antibodies against factors such as CCL5, CXCL1, ICAM-1 by using different antigen retrieval protocols for immunostaining in brain slices. Unfortunately, we could not obtain convincing immunostaining results (e.g., the healthy and LPS-injected mice showed the same background-like staining pattern). We also consulted experts in the field to solve this problem, but we were told it is difficult to immunostain these secreted proteins in vivo due to the lack of reliable antibodies. Nevertheless, previous studies suggest that several proteins such as CXCL1, CXCL10, ICAM-1, LCN-2, MMP3 were mainly produced by reactive astrocytes, and proteins such as CCL-2, CCL-5, IL-1a, CXCL2 were mainly produced by astrocytes and microglia^{19,20}. In the revised manuscript, we further showed the p-STAT3 expression was inhibited in the Adora1 cKO mice, while STAT3 signaling is known to promote the expression of many of the aforementioned cytokines/chemokines (e.g., CXCL10)²¹. These result were highly correlated with the results from astrocytic mRNA^{RiboTag} sequencing, indicating that A1AR-deficient astrocytes reduced the expression of inflammatory-related factors at the protein level.*

14. Including DPCPX or other adenosine receptor antagonists in the behavioral test would provide additional translational relevance of the findings (despite being celltype unspecific).

Author response: *As we explained previously, we already showed evidence that administration of the A1AR antagonist DPCPX at the early phase of systemic inflammation induced by LPS injection could reduce the inflammatory response of microglia (less nuclear p65⁺ microglia), inhibit the expression of inflammation-related cytokines/chemokines, and inhibit the aberrant neuronal activity (less c-Fos expression in neurons) (Fig. 2C; Supplementary Fig. 2d, e, I). In the revised manuscript, we provided additional results demonstrating the DPCPX treatment could ameliorate the behavioural deficits of mice one day post the LPS injection (Supplementary Fig. 10). Therefore, our work highlights the translational potential of A1AR signaling as a therapeutic target for sepsis-induced encephalopathy.*

Reviewer #3 (Remarks to the Author):

I co-reviewed this manuscript with one of the reviewers who provided the listed reports as part of the Nature Communications initiative to facilitate training in peer review and appropriate recognition for co-reviewers.

Author response: *Thank you for your help to improve our work.*

Reviewer #4 (Remarks to the Author):

The manuscript provides a detailed analysis of LPS's early effects on adenosine and astrocytes, with an emphasis on adenosine dynamics following LPS intraperitoneal injections and the molecular, cellular, and behavioral outcomes in cKO mice of A1R in astrocytes. The introduction of new molecular mechanisms in astrocyte responses to systemic inflammation and its implications for mouse phenotypes is both novel and insightful. While recognizing the importance of this research, I have several recommendations for improvement:

Author response: *Thank you for your positive comments on our work as well as for your suggestions to improve the quality of the manuscript.*

Major suggestions:

Interpretations

1. The variation in astrocyte calcium signaling before CPA application between control and cKO groups in Figure S2 needs further analysis and discussion.

Author response: *We provided the analysis of Ca^{2+} signal baselines before CPA applications in the revised version (Supplementary Fig. 4f, g). We also discussed these results accordingly in the Discussion section.*

2. The observed reduction in Ly6B+ neutrophils in cKO mice warrants further exploration to determine if it's due to less BBB impairment or reduced cytokines.

Author response: *The infiltration of peripheral immune cells into the brain parenchyma is a multi-step process including 1) rolling, 2) firm adhesion onto the endothelium, 3) diapedesis across the the endothelium and its basement membrane followed by 4) transmigrating through the glial limitans. Systemic inflammation enhances the focal expression of chemoattractants of peripheral immune cells (e.g., CCL5, CXCL1, ICAM-1) to increase their migration through the endothelium (step 3) and glial limitans (step 4)²². Meanwhile, systemic inflammation also leads to BBB impairment in terms of destroyed tight junctions and basement membranes by increasing inflammation-related proteins such as VCAM-1 and MMPs, which will also facilitate the infiltration of peripheral immune cells²³. Our results demonstrate that astrocytic-A1AR deficiency could*

reduce the global neuroinflammation after LPS injection, leading to the reduction of the chemoattractants of peripheral immune cells as well as the reduction of proteins that impair BBB integrity. Therefore, the reduced infiltration of peripheral immune cells in the Adora1 cKO mice should be attributed to the joint effect of less BBB impairment and reduced chemoattractants. From another point of view, we suggest after LPS injection early reactive astrocytes boost the global neuroinflammation which further provoke BBB dysfunction and induce peripheral immune cell infiltration. We indicate this point in the revised manuscript.

3. In light of recent consensus papers published at Nature Neuroscience in 2021 and at Neuron in 2022, terms like "astrocyte reactivity" and "microglial activation" should be avoided. Focus on specific molecular changes, such as "elevation of xx levels". Please note that Fos family proteins in astrocytes have been shown as downstream elements of the GPCR pathway signaling, but not established as activation markers.

Author response: Thanks for this suggestion. We actually followed the consensus paper about astrocyte at Nat Neurosci which suggests use "astrocyte reactivity" for the "capacity of astrocytes to adopt distinct state(s) in response to diverse pathologies". We also agree to adopt guidelines for definitions of microglia from the consensus paper at Neuron which suggest to use terms like "reactive microglia", "microglial reaction", etc.. In the revised manuscript, we rephrased many terms like "astrocyte activation", "microglial activation" accordingly to be "astrocyte reactivity", "microglia reaction", etc. The details of corrections are tracked in the text.

Clarifications

4. Do GRAB adenosine sensors expressed on astrocyte membranes detect adenosine that is released from, or accumulates on, astrocytes? This inquiry stems from the observed incongruity between the peak of BBB permeability at 6 hours following an LPS i.p. injection and the progressive intensification of GRAB_{Ado1.0} signals from 6 to 24 hours post-injection.

Author response: GRAB_{Ado1.0} is a sensor to detect dynamic changes of the extracellular adenosine, which does not distinguish the sources of extracellular adenosine which can be generated from several routes such as hydrolysis of extracellular ATP by ectonucleotidases, efflux of intracellular adenosine via equilibrative nucleoside transporters (ENTs), etc.. In addition, ATP can be released by all cells via exocytosis or through pannexins; and, pathological stimulation can promote ATP release thereby increasing adenosine levels by hydrolysis. Therefore, it is expected that systemic inflammation-induced aberrant neuronal activity and glial responses could also contribute to the elevated levels of extracellular adenosine after the 6 hpi in terms of releasing more ATP. Indeed, recent studies using GRAB_{ATP1.0} revealed that peripheral LPS injection could evoke extracellular ATP release till at least 24 hpi. These points are discussed in the Discussion

section. Here, we provide an additional figure showing when neuronal activity is inhibited by anesthesia (1.5% isoflurane) the $GRAB_{Ado1.0}$ signal was significantly reduced at 24 hpi, indicating the aberrant neuronal activity contributes to the elevated extracellular adenosine levels (Figure for reviewer). In addition, our current work suggests that during the early phase of systemic inflammation the increased plasma adenosine could directly increase the tone of local extracellular adenosine levels in the brain parenchyma, which correlates with the increased *c-Fos/p-STAT3* expression in astrocytes as well as the peak of BBB impairment (Fig. 3). Therefore, the increased extracellular adenosine in the brain parenchyma after LPS challenge comes from diverse sources during the progression of the pathology, and our results suggest the plasma adenosine contributes to triggering the early astrocyte reactivity by enhancing the local extracellular adenosine.

Figure for reviewer Anesthesia by 1.5% isoflurane reduced the fluorescence intensity of $GRAB_{Ado1.0}$ in the cortex of mice at 24 hpi compared to awake status (Iso0). N=3 mice.

5. The manuscript should explain the rationale behind using K-means clustering, including its specific application and objectives.

Author response: The K-means algorithm was used for unsupervised clustering of gene sets with similarities. Therefore, K-means helps to identify groups of genes with similar expression patterns at 0 hpi, 6 hpi, and 24 hpi, contributing to understanding the change of biological processes at different time points. The main objective is to group genes with similar expression profiles across different time points, which can lead to the discovery of co-regulated genes and pathways as previous studies²⁴ (Figure 4a; Supplementary Fig. 7).

6. The authors are encouraged to provide a more detailed explanation for their choice of brain regions under investigation, particularly the selection of the striatum in Figure 2C and the hippocampus in Figures 5D-F.

Author response: It has been shown that systemic inflammation can induce astrocyte reactivity in the whole brain, however, with region-dependent differences²⁵. Therefore, we attempted to investigate whether A1AR signaling is a common pathway controlling astrocytic responses to systemic inflammation in different brain regions. We initially chose cortex and striatum as representative regions for immunohistochemistry, cytokine expression analysis etc. On the other hand, several prior studies have shown that peripheral LPS injection induces impaired long-term potentiation (LTP) in hippocampus^{24,26-28}. To be comparable with these previous reports, we also chose the well-established protocol for LTP analysis in hippocampus in our experiments. In the revised manuscript, we added new results of immunohistochemical analyses for c-Fos, p-STAT3, p65 expression in the hippocampus to increase the overall compatibility of our results (Fig. 3c, g; Fig. 6c; Fig. 7c).

7. The manuscript mentions having data on A1AR cKO in various cell types but does not present it. Including this data at least to reviewers could be critical in establishing astrocyte specificity.

Author response: In the revised manuscript, we added the results from microglia-A1AR deficient mice and OPC/pericyte-A1AR deficient mice upon challenges by CCPA (A1AR agonist). We found unaltered expression levels of inflammation-related factors in microglia-A1AR deficient mice, which is line with recent transcriptomic profiling studies revealing that A1AR expression in microglia is much lower than in astrocytes and oligodendrocyte lineage cells. In addition, in OPC/pericyte-A1AR mice, we found only Ccl2 expression was inhibited while Ccl5 expression was slightly upregulated, suggesting OPCs/pericytes may contribute to neuroinflammation in response to CCPA (Fig. 2e; Supplementary Fig. 2j, k). However, compared to the effect of inhibiting broader spectrum of inflammation-related factors in astrocyte-A1AR deficient mice (Fig. 2e; Supplementary Fig. 2i), we can still conclude that A1ARs in astrocytes rather than in other glial cell types play a major role in inflammatory response to peripheral CCPA challenge.

Data presentation

8. The authors should double check what the whiskers of each data point mean in Figure 1 B, C and F and define in figure legends.

Author response: Thanks and we clarified this issue in the figure legends as “Summary data are presented as mean \pm SEM in **c, d, g, k**, and as median \pm IQR in **b**”.

9. The layout of data in Figure 2 is challenging to interpret. Using heat plots might enhance clarity for readers.

Author response: Thank you for this suggestion. We used heat plots for these data in the revised version (Fig. 2b-e). To provide the details of data point distributions in each group to our readers, we kept the former bar/dot graphs now in Supplementary Fig. 2.

10. The heatmaps in Figures 3 and S3 lack clarity, particularly regarding what each color represents.

Author response: We improved it by adding the definitions of the scale bars.

11. In Figure 3D, the y-axis labeling in the heatmap is inadequate, making it extremely difficult to identify specific genes. This lack of clarity hinders the interpretation of gene alterations in cKO mice.

Author response: Thanks! We have improved the quality of this panel by adding indication lines.

12. The authors may consider relocating Figure 4G-H panels to follow Figure 3 for a more coherent presentation, as they seem more relevant to it. Also, integrating the analysis of microglial phenotypes with Figure 5 (cellular and behavioral consequences of cKO) could be beneficial.

Author response: Thanks for this suggestion! In the revised manuscript, we moved the cytokine array results (the previous Figure 4G-H) ahead of the microglia results (now as Fig. 5). We feel this rearrangement does improve the coherence of presenting the data. Because we added new data from microglial CD68 expression analysis, now the size of the whole figure depicting microglia reaction is rather big. Therefore, we kept the microglia part as an individual figure, but right before the previous Fig. 5 (now Fig. 6).

Minor points:

1. The authors should ensure to describe “Adora1 cKO”, not just “cKO”, is explicitly defined in all figures and legends for clarity.

Author response: Thanks! We have changed cKO to Adora1 cKO in the text and figures.

2. The authors need to furnish comprehensive details regarding the reagents and equipment used, including catalog numbers and lot numbers where applicable, to enable reproducibility of the results.

Author response: We added the information for all materials in the revised manuscript. In addition, all detailed information is listed in the Supplementary Table 1 of key resources.

3. The references to Nagai et al., 2019a and 2019b seem to be the same. It might be more accurate to cite Nagai et al., 2021 from Neuron.

Author response: Thanks! We have corrected this mistake and also cited Nagai et al., 2021.

References for reviewers:

- 1 Gourine, A. V. *et al.* Release of ATP in the central nervous system during systemic inflammation: real-time measurement in the hypothalamus of conscious rabbits. *J Physiol* **585**, 305-316 (2007). <https://doi.org/10.1113/jphysiol.2007.143933>
- 2 Durkee, C. A. *et al.* *Glia* **67**, 1076-1093 (2019). <https://doi.org/10.1002/glia.23589>
- 3 Vaidyanathan, T. V., Collard, M., Yokoyama, S., Reitman, M. E. & Poskanzer, K. E. Cortical astrocytes independently regulate sleep depth and duration via separate GPCR pathways. *Elife* **10** (2021). <https://doi.org/10.7554/eLife.63329>
- 4 Martin, C., Leone, M., Viviand, X., Ayem, M. L. & Guieu, R. High adenosine plasma concentration as a prognostic index for outcome in patients with septic shock. *Crit Care Med* **28**, 3198-3202 (2000). <https://doi.org/10.1097/00003246-200009000-00014>
- 5 Ramakers, B. P. *et al.* Circulating adenosine increases during human experimental endotoxemia but blockade of its receptor does not influence the immune response and subsequent organ injury. *Crit Care* **15**, R3 (2011). <https://doi.org/10.1186/cc9400>
- 6 Carman, A. J., Mills, J. H., Krenz, A., Kim, D. G. & Bynoe, M. S. Adenosine receptor signaling modulates permeability of the blood-brain barrier. *J Neurosci* **31**, 13272-13280 (2011). <https://doi.org/10.1523/jneurosci.3337-11.2011>
- 7 Hösli, L. *et al.* Direct vascular contact is a hallmark of cerebral astrocytes. *Cell Rep* **39**, 110599 (2022). <https://doi.org/10.1016/j.celrep.2022.110599>
- 8 Klotz, K. N. Adenosine receptors and their ligands. *Naunyn Schmiedebergs Arch Pharmacol* **362**, 382-391 (2000). <https://doi.org/10.1007/s002100000315>
- 9 Sachdeva, S. & Gupta, M. Adenosine and its receptors as therapeutic targets: An overview. *Saudi Pharm J* **21**, 245-253 (2013). <https://doi.org/10.1016/j.jsps.2012.05.011>
- 10 Escartin, C. *et al.* Reactive astrocyte nomenclature, definitions, and future directions. *Nat Neurosci* **24**, 312-325 (2021). <https://doi.org/10.1038/s41593-020-00783-4>
- 11 Yao, Z. *et al.* A high-resolution transcriptomic and spatial atlas of cell types in the whole mouse brain. *Nature* **624**, 317-332 (2023). <https://doi.org/10.1038/s41586-023-06812-z>
- 12 Zhang, M. *et al.* Molecularly defined and spatially resolved cell atlas of the whole mouse brain. *Nature* **624**, 343-354 (2023). <https://doi.org/10.1038/s41586-023-06808-9>
- 13 Kodali, M. C., Chen, H. & Liao, F. F. Temporal unsnarling of brain's acute neuroinflammatory transcriptional profiles reveals panendothelitis as the earliest event preceding microgliosis. *Mol Psychiatry* **26**, 3905-3919 (2021). <https://doi.org/10.1038/s41380-020-00955-5>
- 14 Jonnalagadda, D. *et al.* FTY720 requires vitamin B(12)-TCN2-CD320 signaling in astrocytes to reduce disease in an animal model of multiple sclerosis. *Cell Rep* **42**, 113545 (2023). <https://doi.org/10.1016/j.celrep.2023.113545>
- 15 Groves, A. *et al.* A Functionally Defined In Vivo Astrocyte Population Identified by c-Fos Activation in a Mouse Model of Multiple Sclerosis Modulated by S1P Signaling: Immediate-Early Astrocytes (ieAstrocytes). *eNeuro* **5** (2018). <https://doi.org/10.1523/eneuro.0239-18.2018>
- 16 Nagai, J. *et al.* Hyperactivity with Disrupted Attention by Activation of an Astrocyte Synaptogenic Cue. *Cell* **177**, 1280-1292.e1220 (2019). <https://doi.org/10.1016/j.cell.2019.03.019>

- 17 He, Q., Wang, J. & Hu, H. Illuminating the Activated Brain: Emerging Activity-Dependent Tools to Capture and Control Functional Neural Circuits. *Neurosci Bull* **35**, 369-377 (2019). <https://doi.org/10.1007/s12264-018-0291-x>
- 18 Clayton, B. L. L. *et al.* A phenotypic screening platform for identifying chemical modulators of astrocyte reactivity. *Nat Neurosci* **27**, 656-665 (2024). <https://doi.org/10.1038/s41593-024-01580-z>
- 19 Sofroniew, M. V. Astrocyte Reactivity: Subtypes, States, and Functions in CNS Innate Immunity. *Trends Immunol* **41**, 758-770 (2020). <https://doi.org/10.1016/j.it.2020.07.004>
- 20 Borst, K., Dumas, A. A. & Prinz, M. Microglia: Immune and non-immune functions. *Immunity* **54**, 2194-2208 (2021). <https://doi.org/10.1016/j.immuni.2021.09.014>
- 21 Kim, H. *et al.* Reactive astrocytes transduce inflammation in a blood-brain barrier model through a TNF-STAT3 signaling axis and secretion of alpha 1-antichymotrypsin. *Nat Commun* **13**, 6581 (2022). <https://doi.org/10.1038/s41467-022-34412-4>
- 22 Galea, I. The blood-brain barrier in systemic infection and inflammation. *Cell Mol Immunol* **18**, 2489-2501 (2021). <https://doi.org/10.1038/s41423-021-00757-x>
- 23 Paolicelli, R. C. *et al.* Microglia states and nomenclature: A field at its crossroads. *Neuron* **110**, 3458-3483 (2022). <https://doi.org/10.1016/j.neuron.2022.10.020>
- 24 Shemer, A. *et al.* Interleukin-10 Prevents Pathological Microglia Hyperactivation following Peripheral Endotoxin Challenge. *Immunity* **53**, 1033-1049.e1037 (2020). <https://doi.org/10.1016/j.immuni.2020.09.018>
- 25 Hasel, P., Rose, I. V. L., Sadick, J. S., Kim, R. D. & Liddelow, S. A. Neuroinflammatory astrocyte subtypes in the mouse brain. *Nat Neurosci* **24**, 1475-1487 (2021). <https://doi.org/10.1038/s41593-021-00905-6>
- 26 Izumi, Y. *et al.* A Proinflammatory Stimulus Disrupts Hippocampal Plasticity and Learning via Microglial Activation and 25-Hydroxycholesterol. *J Neurosci* **41**, 10054-10064 (2021). <https://doi.org/10.1523/jneurosci.1502-21.2021>
- 27 de Deus, J. L. *et al.* Inhaled molecular hydrogen reduces hippocampal neuroinflammation, glial reactivity and ameliorates memory impairment during systemic inflammation. *Brain Behav Immun Health* **31**, 100654 (2023). <https://doi.org/10.1016/j.bbih.2023.100654>
- 28 Lonnemann, N. *et al.* IL-37 expression reduces acute and chronic neuroinflammation and rescues cognitive impairment in an Alzheimer's disease mouse model. *Elife* **11** (2022). <https://doi.org/10.7554/eLife.75889>

REVIEWERS' COMMENTS

Reviewer #1 (Remarks to the Author):

All my questions and criticism was adequately taken care of.

Reviewer #2 (Remarks to the Author):

This is the revised version of a manuscript concerning the role of Adenosine in sepsis-associated encephalopathy.

In the rebuttal letter, the authors addressed the points brought forward by this reviewer in sufficient extent and have implemented novel datasets into the revised version of the manuscript.

Thus, I have no additional questions and suggest to accept the manuscript in its current form, if the other reviewers agree as well.

Reviewer #4 (Remarks to the Author):

Overall, the authors have carefully addressed the points I raised. The discussion is well-articulated, and the added results from the microglia-A1AR deficient mice and OPC/pericyte-A1AR deficient mice are of significant importance.

I have one minor suggestion:

The heat maps in Figure 2 effectively summarize the data that was previously presented in complex bar graphs, which is a positive change. However, it would be very helpful for readers if the groups being compared were clearly labeled (such like xxx vs yyy). The current version may confuse readers regarding which groups serve as controls.

Reviewer #1 (Remarks to the Author):

All my questions and criticism was adequately taken care of.

Author response: *Thank you for your positive comments.*

Reviewer #2 (Remarks to the Author):

This is the revised version of a manuscript concerning the role of Adenosine in sepsis-associated encephalopathy.

In the rebuttal letter, the authors addressed the points brought forward by this reviewer in sufficient extent and have implemented novel datasets into the revised version of the manuscript.

Thus, I have no additional questions and suggest to accept the manuscript in its current form, if the other reviewers agree as well.

Author response: *Thank you for your positive comments.*

Reviewer #4 (Remarks to the Author):

Overall, the authors have carefully addressed the points I raised. The discussion is well-articulated, and the added results from the microglia-A1AR deficient mice and OPC/pericyte-A1AR deficient mice are of significant importance.

Author response: *Thank you for your positive comments.*

I have one minor suggestion:

The heat maps in Figure 2 effectively summarize the data that was previously presented in complex bar graphs, which is a positive change. However, it would be very helpful for readers if the groups being compared were clearly labeled (such like xxx vs yyy). The current version may confuse readers regarding which groups serve as controls.

Author response: *Thank you for this suggestion. We have added this information in the figure.*